# Enhanced upward motion through the troposphere over the tropical western Pacific and its implications for the transport of trace gases from the troposphere to the stratosphere

Kai Qie [1], Wuke Wang[2*], Wenshou Tian [1], Rui Huang[1], Mian Xu[1], Tao Wang[1], Yifeng Peng[1]

[1]*College of Atmospheric Sciences, Lanzhou University, Lanzhou 730000, China*

[2] *Department of Atmospheric Science, China University of Geosciences, Wuhan 430074, China*

*Corresponding author: Wuke Wang (wangwuke@cug.edu.cn)

**Abstract**

The tropical western Pacific (TWP) is a preferential area of air uplifting from the surface to the upper troposphere. A significantly intensified upward motion through the troposphere over the TWP in the boreal wintertime (November to March of the next year, NDJFM) has been detected using multiple reanalysis datasets. The upward motion over the TWP is intensified at rates of $8.0\pm3.1\%$ decade$^{-1}$ and $3.6\pm3.3\%$ decade$^{-1}$ in NDJFM at 150 hPa from 1958 to 2017 using JRA55 and ERA5 reanalysis datasets, while the MERRA2 reanalysis data show a $7.5\pm7.1\%$ decade$^{-1}$ intensified upward motion for the period 1980-2017. Model simulations using the Whole Atmosphere Community Climate Model, version 4 (WACCM4) suggest that warming global sea surface temperatures (SSTs), particularly SSTs over the eastern maritime continent and tropical western Pacific, play a dominant role in the intensification of the upward motion by strengthening the Pacific Walker circulation and enhancing the deep convection over the TWP. Using CO as a tropospheric tracer, the WACCM4 simulations show that an increase of CO at a rate of 0.4 ppbv decade$^{-1}$ at the layer 150-70 hPa in the tropics is mainly resulted from the global SST warming and the subsequent enhanced upward motion over the TWP in the troposphere and strengthened tropical upwelling of Brewer-Dobson (BD) circulation in the lower stratosphere. This implies that more tropospheric trace gases and aerosols from both natural maritime source and outflow from polluted air from South Asia may enter the stratosphere through the TWP region and affect the stratospheric chemistry and climate.

**Keywords**: Upward motion; Troposphere-to-stratosphere transport; Tropical western

Pacific; Trend; Sea surface temperature

## 1 Introduction

The tropical western Pacific (TWP) is a critical region for tropical and global climate (e.g., Webster et al., 1996; Hu et al., 2020). It has the largest area of warm sea surface temperature (exceeding 28 $^{\circ}$C) which fuels intense and massive deep convection and thus is the largest source of latent heat and water vapor into the atmosphere (Webster and Lukas, 1992). The TWP region is also the most important source of tropospheric air entering the stratosphere due to the strong upward motion and deep convection over this region (e.g., Fueglistaler et al., 2004; Pan et al., 2016). Through the TWP region, tropospheric trace gases, e.g., the natural maritime bromine-containing substances and outflow from anthropogenic emissions from South Asia, are lifted to the upper troposphere and lower stratosphere (UTLS) by the strong upward motion and the deep convection and subsequently into the stratosphere by the large-scale upwelling (e.g., Levine et al., 2007, 2008; Navarro et al., 2015), which affect the ozone concentration and other chemical processes in the stratosphere (e.g., Feng et al., 2007; Sinnhuber et al., 2009). At the same time, the TWP region has the lowest cold-point tropopause temperature (CPTT) over the globe and plays an important role in controlling the water vapor concentration in the stratosphere (e.g., Fueglistaler et al., 2009; Newell and Gould-Steward, 1981; Pan et al., 2016; Randel and Jensen, 2013). The TWP is an important region for tropospheric trace gases being transported from the troposphere to the stratosphere, and therefore influencing the stratospheric chemistry (e.g., Fueglistaler et al., 2004; Levine et al., 2007; Krüger et

al., 2008; Pan et al., 2016).

The TWP was thought to be the main pathway of the troposphere-to-stratosphere

transport. A concept of "stratospheric fountain" was proposed by Newell and
Gould-Steward (1981), which suggested that the poor-water vapor air in the
stratosphere stems mainly from the TWP region. However, following studies using the
observational and reanalysis data showed that there is subsidence at the
near-tropopause level over the maritime continent, which is named as the
"stratospheric drain" (Gettelman et al., 2000; Sherwood, 2000; Fueglistaler et al.,
2004). Further studies verified that the large-scale transport from the tropical
tropopause layer (TTL) to the stratosphere is dominated by the upward branch of the
Brewer-Dobson (BD) circulation (Brewer, 1949; Dobson, 1956; Holton et al., 1995)
while the local upwelling may play a minor role (e.g., Levine et al., 2007; Fueglistaler
et al., 2009; Schoeberl et al., 2018).

Though the vertical transport from TTL to the lower stratosphere is dominated by

the BD circulation, numerous studies confirmed that the TWP region is an important
pathway of the surface air entering the TTL (Fueglistaler et al., 2004; Levine et al.,
2007; Krüger et al., 2008; Haines and Esler, 2014). Based on a trajectory model,
Fueglistaler et al. (2004) pointed out that approximately 80% of the trajectories
ascending into the stratosphere from the TTL are originated from the TWP region.
Bergman et al. (2012) suggested that the tropospheric air over the TWP enters the
stratosphere mainly in boreal winter, while less air over the TWP could be transported
into the stratosphere during boreal summer. Other studies also found that the TWP
region is an important source of the tropospheric trace gases in the TTL (e.g., Newton
et al., 2018; Pan et al., 2016; Wales et al., 2018), even the polluted air from East Asia
could be transported rapidly to Southeast Asia by meridional winds and subsequently
be elevated to the tropical upper troposphere by the strong upward motion and the
deep convection (Ashfold et al., 2015). Hence, the strength of the upward motion over
the TWP region during boreal winter is a key feature for understanding the variations
of trace gases in the TTL and therefore important for stratospheric chemistry and
climate.
The strength of the TWP upward motion is closely related to atmospheric
circulation and deep convection. The ascending branch of the Pacific Walker
circulation and the strong deep convection over the TWP allow rapid transport from
the surface to the upper troposphere (Hosking et al., 2012). In association with global
warming, atmospheric circulation, deep convection as well as the boundary conditions
(e.g., sea surface temperature; SST) have been changed. For example, the Hadley cell
has been extended to the subtropics and the Walker circulation over the Pacific has
been shifted westward over the past decades (e.g., Lu et al., 2007; Garfinkel et al.,
2015; Ma and Zhou, 2016). At the same time, SSTs over most of areas are getting
warmer (Cane et al., 1997; Deser et al., 2010), which modulates the deep convection
and atmospheric wave activities in the troposphere and then lead to changes of
atmospheric circulations from the troposphere and the stratosphere (e.g., Garfinkel et
al., 2013; Xie et al., 2012, 2014a; Wang et al., 2015; Hu et al., 2016; Lu et al., 2020).
However, how the strength of the upward motion in the lower TTL over the TWP
region has been changed over the past decades remains unclear. In this study, we
investigate the long-term trend of the upward motion over the TWP using JRA55,
ERA5, and MERRA2 reanalysis datasets and different WACCM4 simulations as
described in Section 2. The implication of the changes in the upward motion over the
TWP to the transport of trace gases from the surface to the UTLS will be discussed in
Section 3.
**2 Data and method**
**Reanalysis data.** To investigate the long-term trend of the upward motion over
the TWP through the troposphere, three most recent reanalysis datasets, including
JRA55 from the Japan Meteorological Agency (JMA), ERA5 from the European
Centre for Medium Range Weather Forecasting (ECMWF) and MERRA2 from the
National Aeronautics and Space Administration/Global Modeling and Assimilation
Office (NASA/GMAO) are used in this study (Table 1). The JRA55 data, covering the
period from 1958 to present, are interpolated to the standard pressure levels and
$1.25° \times 1.25°$ horizontal mesh (Harada et al. 2016). The ERA5 reanalysis is the newest
generation product from the ECMWF (Hersbach et al., 2020). The ERA5 data are
based on the Integrated Forecasting System (IFS) Cy41r2, which includes the
improved model physics, core dynamics and data assimilation. The ERA5 data also
extend back to 1958, which is coinciding with the time that radiosonde observations
in the Arctic became more systematic and regular. It should be noted that the ERA5
data suffer from a bias during 2000-2006, and are replaced by the ERA5.1 data in this
period here. The MERRA2 data are also used (Gelaro et al., 2017), which are
produced by NASA/GMAO using Goddard Earth Observing System model (GEOS).
Although the horizontal and vertical resolution of MERRA2 data are similar to
MERRA data, the MERRA2 data represent UTLS processes better (Gelaro et al.,
2017). The monthly mean air temperature, horizontal wind fields and vertical velocity
at different pressure levels are extracted from the three Reanalysis datasets. In the
present study, we mainly focus on the upward motion over TWP region in NDJFM,
which is defined as 20°S-10°N, 100°E-180° due to the strong upward motion (Fig. 1)
and significantly increasing trends of the upward motion (Fig. 2) over there.
A special caution is needed because of the limitations of reanalysis data. The
reanalysis datasets assimilate observational data based on the ground- and
space-based remote sensing platforms to provide more realistic data products.
However, previous studies suggested that there are still uncertainties in the reanalysis
data (e.g., Simmons et al., 2014; Long et al., 2017; Uma et al., 2021). The accuracy of
the vertical velocity in reanalysis data sets has been evaluated by the Reanalysis
Intercomparison Project (Fujiwara et al., 2017), which is initiated by the
Stratosphere-troposphere Processes And their Role in Climate (SPARC). Results of a
comparison between the radar observed data and the reanalysis data indicate that the
updrafts in the UTLS are captured well near the TWP even though there are still large
biases in the reanalysis datasets and the updrafts from the JRA55 data are stronger
than those from the ERA5 and MERRA2 data (Uma et al. 2021). Additionally,
discontinuities in the reanalysis data due to different observing systems (for example,
transition from TOVS to ATOVS) may still exist (e.g., Long et al., 2017), which could
lead to uncertainties in the long-term trend of a certain meteorological field.
Hitchcock (2019) suggested that the reanalysis uncertainty is larger in the radiosonde
era (after 1958) than in the satellite era (after 1979), but the radiosonde era is of
equivalent value to the satellite era because the dynamical uncertainty dominates in
the both eras. The data in the radiosonde era (1958-1978) used in the present study
may induce uncertainties in our results. Therefore, we discuss the trends for both the
periods of 1958-2017 and 1980-2017. In addition, we combine three most recent
reanalysis datasets (JRA55, ERA5, and MERRA2) to obtain relatively robust results.

**Observed CO data.** Since CO has a photochemical lifetime in the range of 2-3

months (Xiao et al., 2007), it could be utilized as a tracer of cross-region transport in
the troposphere and the lower stratosphere (Park et al., 2009). Here, CO is used as a
tropospheric tracer to indicate the vertical transport from the near-surface to the upper
troposphere and the lower stratosphere. The CO data used in the present study are
from space-borne Microwave Limb Sounder (MLS; Livesey et al., 2015) observation
and Measurements Of Pollution In The Troposphere instrument (MOPITT; Deeter et
al., 2019). MLS is carried by Aura, which has a sun-synchronous orbit at 705 km with
a 16-day repeat cycle. MLS observations are made from 82°S to 82°N and cover the
period from 2005 to the present. MLS provides the CO data from the upper
troposphere to the mesosphere. MLS CO v4 level1 data used in the present study are
processed using the recommended procedures (Livesey et al., 2015) and interpolated
into a 5°×5° horizontal mesh. MOPITT CO data are also used for comparison.
MOPITT instrument is aboard on the Terra satellite permitting retrievals of CO
vertical profiles using both thermal-infrared and near-infrared measurements and has
a field of view of 22 km×22 km. The Terra satellite was launched in 1999 with a 705
km sun-synchronous orbit. MOPITT provides the CO data from the surface to the
upper troposphere during the period of 2000/03 to the present. Here, we use the
daytime only MOPITT v8 level3 CO data. For comparison, we focus on the CO
concentrations in MLS and MOPITT data at similar level (215 hPa in MLS data and
200 hPa in MOPITT data, respectively).
**SST and outgoing longwave radiation (OLR) data.** SST data are used in this
study to investigate the relationship between the upward motion and SSTs. The SST
data are from the HadISST dataset (1°×1° horizontal mesh) during 1958-2018
(Rayner et al., 2003). OLR is often utilized to reflect the deep convection in the
tropics. The OLR data are extracted from NOAA Interpolated OLR dataset on a
2.5°×2.5° horizontal mesh during 1974/11-2018/03 (Liebmann and Smith, 1996).
**Model simulations.** A series of model simulations with the Whole Atmosphere
Community Climate Model version 4 (WACCM4) are performed to find out the main
impact factors of the trend of the upward motion over the TWP (Table 2). The
WACCM4 is a chemical-climate model with a horizontal resolution of 1.9°×2.5°
(Marsh et al., 2013). The WACCM4 has vertical 66 levels from the surface to 145 km
with vertical resolution of approximately 1 km in the UTLS, which is numerously
used to investigate the transport of the trace gases from the troposphere to the
stratosphere (e.g., Randel et al., 2010; Xie et al., 2014b; Minganti et al., 2020). A
hindcast simulation (Control simulation) is performed with observed greenhouse
gases, solar irradiances, and prescribed SSTs (HadISST dataset is used) during
1955-2018. A single-factor controlling simulation (Fixsst simulation) is done for the
same period with the same forcings, except that the global SSTs are fixed to the
climatological mean values during 1955-2018 (long-term mean for each calendar
month during 1955-2018).
To figure out the impact of the warming SST over the TWP region on the
intensification of the upward motion over the TWP region, a couple of time-slice
simulations (R1 and R2) are also integrated for 33 years. The SSTs over the eastern
maritime continent and tropical western Pacific (20°S-20°N, 120°E-160°E) in the
boreal wintertime (November to March of the next year, NDJFM) in R1 are
prescribed as the climatological mean SSTs during 1998-2017, while the SSTs over
other regions are fixed as the climatological mean SSTs during 1958-2017. The SSTs
in R2 are the same as the SSTs in R1 except that the SSTs over the region (20°S-20°N,
120°E-160°E) in NDJFM are prescribed as the climatological mean SSTs during
1958-1977. Since the SSTs over the eastern maritime continent and tropical western
Pacific (20°S-20°N, 120°E-160°E) show significantly warming trends, the SSTs
during 1998-2017 are higher than the SSTs during 1958-1977 (approximately 0.5 K).
Hence, the difference between R1 and R2 reflects the impact of the warmed SSTs
over the eastern maritime continent and tropical western Pacific (20°S-20°N,
120°E-160°E) on the atmospheric circulation. The first 3 years of the numeric
simulations are not used in the present study to provide a spin-up.
**Transformed Eulerian Mean (TEM) calculation.** To diagnose the changes in
the BD circulation, the meridional and vertical velocities of the BD circulation are
calculated by the TEM equations (Andrews and Mcintyre, 1976):
$$v^* = \bar{v} - \frac{1}{\rho}\left(\frac{\overline{\rho v'\theta'}}{\theta_z}\right)_z$$
$$w^* = \bar{w} + \frac{1}{a\cos\varphi}\left(\cos\varphi\frac{\overline{v'\theta'}}{\theta_z}\right)_\varphi$$
Where $v^*$ and $w^*$ denote the meridional and vertical velocities of the BD circulation;
the overbar represents the zonal mean; the prime denotes the deviation from the zonal
mean; $\theta$, $a$, $\varphi$, and $\rho$ indicate the potential temperature, the radius of the earth,
the latitude, and the standard density.
**Linear trends and the significance test**. The linear trends are estimated using a
simple least square regression method. The significances of the correlation
coefficients, mean differences, and trends are determined via a two-tail Student's t-test.
The confidence interval of trend is calculated using the following equation (Shirley et
al., 2004): $\left(b - t_{1-\frac{\alpha}{2}}(n-2)\sigma_b, b + t_{1-\frac{\alpha}{2}}(n-2)\sigma\right)$
where b is the estimated slope, $\sigma$ denotes the standard error of the slope, and
$t_{1-\frac{\alpha}{2}}(n-2)$ represents the value of t-distribution with the degree of freedom equal to
*n*-2. α is the two-tailed confidence level. $\sigma$ is calculated as: $\sigma = b\sqrt{\frac{\frac{1}{r^2}-1}{n-2}}$.
**3 Results**
**3.1 Enhanced upward motion over the TWP**
According to previous studies, the lapse-rate tropopause is a good proxy to
separate the tropospheric and the stratospheric dynamic behavior (vertical motion
dominated and horizontal mixing dominated, respectively) over the TWP (Pan et al.,
2019). Since the lapse-rate tropopause over the TWP in the boreal winter is near 100
hPa (not shown), we utilize the vertical velocity at 150 hPa to reflect the vertical
transport in the upper troposphere. Figure 1 shows mean values of the vertical
velocity at 150 hPa for each month averaged over 60 years from 1958 to 2017. The
TWP region at the UTLS level has strong upward motion due to the frequent intense
deep convection and the Pacific Walker circulation. It is noteworthy that there is
strong upward motion at 150 hPa in NDJFM over the TWP, while the upward motion
in other months shifts northward corresponding to the Asia summer monsoon. This is
consistent with previous studies (Newell and Gould-Steward, 1981; Bergman et al.,
2012). Therefore, we mainly focus on the changes in the upward motion in NDJFM,
which is more important to the transport of air over the TWP from the lower
troposphere to the TTL compared to the summer months (as shown in Fig. 1) and
subsequently to the lower stratosphere. As seen in Figs. 1a-c and 1k-l, the upward
motion ($w$) at 150 hPa is most evident over the region 20°S-10°N, 100°E-180°, which
is used to indicate the TWP in the following analysis. The climatological mean 150
hPa vertical velocity ($w$) in NDJFM in ERA5 during 1958-2017 and MERRA2 during
1980-2017 are also given in Supplementary Fig. 1. Comparing with the 150 hPa $w$ in
NDJFM using JRA55, the 150 hPa $w$ in ERA5 and MERRA2 data shows larger
values (maximum larger than 1.5 cm s$^{-1}$) over the land areas but smaller values
(minimum less than -0.4 cm s$^{-1}$) over the marine area. Notably, the 150 hPa $w$ shows
no subsidence over the maritime continent, while there is descending motion over the
maritime continent at 100 hPa (Supplementary Fig. 2), which is referred to the
"stratospheric drain" (Gettleman et al., 2000; Sherwood, 2000).

Figure 2 displays the linear trends of $w$ in the upper (150 hPa), middle (500 hPa)

and lower (700 hPa) troposphere in NDJFM from 1958 to 2017 using JRA55, ERA5,
and MERRA2 reanalysis datasets. The 150 hPa $w$ increased significantly over most
areas of the TWP during 1958-2017 (Fig. 2). At the same time, the upward motion
over the TWP in the lower and middle troposphere also mainly shows positive trends
(Figs. 2d and g). This indicates that the upward motion over the TWP is increasing
through the troposphere from 1958 to 2017. Such an enhancement of the upward
motion over the TWP is evident in all three reanalysis datasets used here (JRA55,
ERA5, and MERRA2), although there are also some differences between the three
reanalysis datasets. For example, the trends of the horizontal winds in the upper
troposphere in MERRA2 (Fig. 2c) are larger than those in JRA55 and ERA5 (Figs. 2a
and b). There are negative trends of vertical velocity in JRA55 and ERA5 while
positive trends of vertical velocity in MERRA2 over the northern Pacific. However,
these differences are mainly due to the different time periods which are used to
calculate the linear trends in JRA55 (1958-2017), ERA5 (1958-2017) and MERRA2
(1980-2017). Supplementary Fig. 3 gives the trends of $w$ and horizontal winds in
NDJFM during 1980-2017 derived from JRA55, ERA5, and MERRA2 data, which
shows insignificant differences between these reanalysis datasets. The trend patterns
of the horizontal winds in JRA55, ERA5, and MERRA2 are consistent with each
other (Supplementary Fig. 3). For the trends of vertical velocity, significantly positive
trends over the TWP region can be noted in the JRA55, ERA5, and MERRA2 datasets,
although the trends in ERA5 are slightly weaker than those in JRA55 and MERRA2
(Fig. 2 and Supplementary Fig. 3). Comparing to the negative trends of the vertical
velocity over the central Pacific in JRA55 and ERA5, the negative trends in MERRA2
extend more northward (Supplementary Fig. 3).
The time series of the upward motion intensity over the TWP from different
datasets are given in Fig. 3. The intensity of the upward motion over the TWP used in
Fig. 3 is simply defined as the area-averaged upward mass flux at a specific level, and
the standardized intensity is calculated as the intensity divided by the standard
deviation of the intensity at the corresponding level. The intensity of the upward
motion over the TWP at 150 hPa increased significantly in NDJFM during last
decades, which can be confirmed by all the three reanalysis datasets (Fig. 3). The
intensity of the upward motion over the TWP at 150 hPa increased $3.0\pm1.2\times10^9$ kg s$^{-1}$
decade$^{-1}$ ($8.0\pm3.1\%$ decade$^{-1}$), $1.3\pm1.2\times10^9$ kg s$^{-1}$ decade$^{-1}$ ($3.6\pm3.3\%$ decade$^{-1}$), and
$3.0\pm2.8\times10^9$ kg s$^{-1}$ decade$^{-1}$ ($7.5\pm7.1\%$ decade$^{-1}$) in JRA55, ERA5, and MERRA2 data,
respectively (Table 3). As shown in Figs. 3b and c, the intensity of the upward motion
at 500 hPa and 700 hPa in JRA55 and the intensity of the upward motion at 500 hPa
in ERA5 over the TWP also increased significantly at 95% confidence level
($4.6\pm2.6\times10^9$ kg s$^{-1}$ decade$^{-1}$, $2.9\pm1.7\times10^9$ kg s$^{-1}$ decade$^{-1}$, and $2.5\pm2.5\times10^9$ kg s$^{-1}$
decade$^{-1}$, respectively). The increasing trends of the intensity of the upward motion at
700 hPa in ERA5 and at 500 hPa and 700 hPa in MERRA2 are significant at the 90%
confidence level at rates of $1.9\pm1.6\times10^9$ kg s$^{-1}$ decade$^{-1}$, $5.4\pm5.3\times10^9$ kg s$^{-1}$ decade$^{-1}$
and $3.9\pm3.8\times10^9$ kg s$^{-1}$ decade$^{-1}$, respectively. This suggests a comprehensive
enhancement of vertical velocity through the whole troposphere, which is evident
from the surface to 100 hPa (Supplementary Fig. 4). It can also be inferred that the
upward motions over the TWP increased at different rates during the past decades due
to the difference between JRA55, ERA5, and MERRA2 data. Hence, caution is
suggested when investigating the trend of the upward motion over the TWP using the
reanalysis data. While the trace gases in the TTL are modulated by the upward motion
and subsequent vertical transport (e.g., Garfinkal et al., 2013; Xie et al., 2014b), such
a strengthening of the upward motion over the TWP may lead to more tropospheric
trace gases in the TTL.
The changes in the atmospheric circulation at the UTLS level in the tropics are
closely related to the changes in the tropical deep convection and SSTs (e.g., Levine
et al., 2008; Garfinkal et al., 2013; Xie et al., 2020). Here, the trends of observed OLR
provided by NOAA (see Section 2) in NDJFM during 1974-2017 are shown in Fig. 4a.
Though the time period of the observed OLR data is shorter than the time period we
analyzed, the changes in OLR could partly reflect the changes in the deep convection
during 1958-2017. The OLR shows significantly negative trends over the TWP which
indicates intensified deep convection over the TWP. The OLR trend pattern is very
similar to the trend pattern of the 150 hPa $w$ (Figs. 2a-c), which indicates that the
increasing trends of 150 hPa $w$ are closely related to the intensified deep convection
over the TWP. The intensified deep convection not only lead to the strengthened
upward motion in the UTLS (Highwood and Hoskins, 1998; Ryu and Lee, 2010), but
also result in the decreased temperature near the tropopause which plays a dominant
role in modulating the lower stratospheric water vapor concentration (e.g., Hu et al.,
2016; Wang et al., 2016). Corresponding to the enhanced deep convection over the
TWP, the CPTT derived from JRA55 data (see Fig. 4b) shows significantly decreasing
trends over the TWP in NDJFM during 1958-2017, which is consistent with Xie et al.,
(2014a). However, negative trends are also found in other regions in low and
mid-latitudes, except over the central and east Pacific. It should be noted that the
CPTT from different reanalysis datasets may show different trends even for the
satellite period (Tegtmeier et al., 2020). Additionally, the JRA55 data before 1978
may also lead to uncertainties in the CPTT trends. Caution is needed when discussing
the trends of CPTT from reanalysis datasets.
The changes in the deep convection over the tropical Pacific may be related to
the changes in the Pacific Walker circulation. The Pacific Walker circulation shows a
significant intensification over the past decades (e.g., Meng et al., 2012; L'Heureux et
al., 2013; McGregor et al., 2014). The vertical velocity at 500 hPa and 150 hPa shows
significantly positive trends over the TWP in NDJFM during 1958-2017 (Fig. 2).
Meanwhile, the lower tropospheric zonal wind shows easterly trends over the tropical
Pacific, while the upper tropospheric zonal wind shows westerly trends over the
tropical Pacific, which suggests a strengthened Pacific Walker circulation and is
consistent with previous studies (Hu et al., 2016; Ma and Zhou, 2016).
The strengthened Pacific Walker circulation is closely related to the changes in
the SSTs (e.g., Meng et al., 2012; Ma and Zhou, 2016). The trends of the SSTs in

NDJFM during 1958-2017 are shown in Fig. 4c. The SST shows significantly

warming trends almost over the world except the central Pacific in NDJFM during

1958-2017. In addition, the intensity of the upward motion over the TWP is

significantly correlated with the SST (Fig. 4d), which suggests that the SST has

important effects on the upward motion over the TWP. The correlation coefficient in

Fig. 4d shows a La Niña-like pattern and indicates that the ENSO events exert

important impacts on the upward motion over the TWP (Levine et al., 2008). The

SSTs over the TWP are mainly positively correlated with the upward motion intensity

over the TWP with negative correlations shown over the western maritime continent,

while the SSTs over tropical central, eastern Pacific, and Indian Ocean show negative

correlations with the intensity of the upward motion over the TWP. The SSTs over the

Atlantic Ocean are poorly correlated with the upward motion intensity over the TWP

(not shown). This result suggests that the changes in global SSTs may be the primary

driver of the strengthened Pacific Walker circulation, which leads to enhanced deep

convection and intensified upward motion over the TWP.

It could be found that there are extreme minima (1982, 1991, and 1997) in Fig. 3,

which may be related to the El Niño events occurred in these years. To further figure

out the impact of ENSO events on the upward motion over the TWP, Supplementary

Fig. 5 displays the time series of the standardized intensity of the upward motion over

the TWP at 150 hPa, 500 hPa, and 700 hPa in NDJFM in JRA55, ERA5, and

MERRA2 with the ENSO signal removed using the linear regression method (Hu et

al., 2018; Qie et al., 2021). The extreme minima (1982, 1991, and 1997) become

much weaker in Supplementary Fig. 5 than those in Fig. 3, which indicates that the El Niño events are responsible for the extreme minima. The upward motions over the TWP at 150 hPa, 500 hPa, and 700 hPa in NDJFM in JRA55, ERA5, and MERRA2 still show statistically significant increasing trends after removing the ENSO signal in Supplementary Fig. 5, which suggests that ENSO events exert limited impacts on the trends of the upward motion over the TWP in NDJFM during 1958-2017.

**3.2 Simulated trend of the upward motion over the TWP and its potential mechanism**

To verify the impact of SST on the trend of the upward motion over the TWP, a couple of model simulations with WACCM4 are employed in the following analysis. Consistent with the results shown using the reanalysis data (Figs. 2a-c), the simulated 150 hPa $w$ (Control simulation) shows significantly increasing trends over the TWP and decreasing trends over the tropical eastern Pacific in NDJFM during 1958-2017 (Fig. 5a). Additionally, the 150 hPa $w$ simulated in the Fixsst simulation shows weak trends over the TWP (Fig. 5b). The difference between the Control and the Fixsst simulations suggests that the trends of the 150 hPa $w$ over the TWP region is dominated by the changes in the global SSTs during 1958-2017. There are also significantly positive trends of the vertical velocity over the TWP in the lower (700 hPa) and middle troposphere (500 hPa) in the Control simulation, while the zonal winds are also enhanced over the tropical Pacific. The vertical velocity over the TWP in the Fixsst simulation shows weak negative trends and the changes in zonal winds over the tropical Pacific are very weak. This confirms the dominant role of the

changes in global SSTs on the enhancement of the Walker circulation.

Previous studies found that the changes in the intensity of the Pacific Walker

circulation and the stratospheric residual circulation are closely related to the changes
in tropical SST (Meng et al., 2012; Tokinaga et al., 2012; Lin et al., 2015). As
suggested by the correlation coefficients between the upward motion at 150 hPa over
the TWP and SSTs in Fig. 4d, warmer SSTs over the tropical central and eastern
Pacific, and Indian Ocean may lead to a weakened upward motion over the TWP
(negative correlation). The warming trends of SSTs over the eastern maritime
continent and tropical western Pacific may result in an intensification of the upward
motion over the TWP. To verify the impact of the changes in the SSTs over eastern
maritime continent and tropical western Pacific on the trends of the upward motion
over the TWP, a couple of single-factor controlling time-slice simulations (R1 and R2)
are performed with only SSTs over eastern maritime continent and tropical western
Pacific (20°S-20°N, 120°E-160°E) in NDJFM changed in these two simulations. In
R1, the SSTs over the eastern maritime continent and tropical western Pacific are
prescribed as the climatological mean SSTs during 1958-2017, while the SSTs over
the eastern maritime continent and tropical western Pacific in R2 are prescribed as the
climatological mean SSTs during 1958-1977 (more details are given in the section 2).
The differences of the wind fields between R1 and R2 are shown in Fig. 6. The 150
hPa $w$ shows significantly positive anomalies over the TWP and negative anomalies
over the tropical eastern Pacific, which is consistent with the trends of the 150 hPa $w$
in the Control simulation and the reanalysis datasets (Figs. 2 and 5). The upward mass
flux over the TWP at 150 hPa increased approximately 27% in the R1 comparing with
R2 due to the warming SSTs over the eastern maritime continent and tropical western
Pacific (approximately 0.5 K). The upward motion in the lower and middle
troposphere over the TWP shows increasing trends due to the enhanced convergence
induced by the warmer SSTs over the TWP. This result is consistent with Hu et al.
(2016), which suggested that the increased zonal gradient of the SSTs over the
tropical Pacific could lead to a strengthened Pacific Walker circulation and an
enhanced upward motion over the TWP. Therefore, the warmer SSTs over the TWP
could contribute largely to the trend of the upward motion over the TWP in NDJFM
during 1958-2017.
The changes in the OLR simulated in WACCM4 associated with the changes in
the global SSTs are shown in Fig. 7. There are significantly enhanced deep convection
as indicated by OLR over the TWP due to the strengthened convergence in the
Control simulation, while the deep convection shows weak and even decreasing
trends over the TWP in the Fixsst simulation (Figs. 7a and b). The enhanced deep
convection over the TWP could lead to the enhancing trends of the upward motion.
Hence, it can be inferred that the changes in the global SSTs are responsible for the
intensification of the Pacific Walker circulation, and the enhanced deep convection
and a stronger upward motion over the TWP which could extend to the upper
troposphere.
**3.3 Implications for the concentrations of water vapor and CO in the TTL**
**and lower stratosphere**
Previous studies showed that the enhanced deep convection and upward motion
could lead to increased CO in the UTLS (e.g., Duncan et al., 2007; Livesey et al.,
2013). At the same time, water vapor mixing ratios in the UTLS may increase due to
the enhanced upward motion which could bring more wet air from low altitude to
high altitude (e.g., Rosenlof, 2003; Lu et al., 2020). However, the water vapor mixing
ratios in the lower stratosphere also depend on the tropopause temperature (e.g.,
Highwood and Hoskins, 1998; Garfinkel et al., 2018; Pan et al., 2019). Hence, the
relationship between the intensity of upward motion and the water vapor
concentration in the UTLS is complex. Here, the relationship between the trends of
the upward motion over the TWP and the changes in CO and water vapor in the ULTS
simulated with WACCM4 are analyzed.
The trends of CPTT, the 100 hPa streamfunction, and the water vapor
concentration are shown based on the Control and the Fixsst simulation as well as
their difference in Figs. 7d-i. The changes in the deep convection could lead to the
changes in the atmospheric circulation by releasing the latent heat. The changes in the
tropical deep convection lead to a Rossby-Kelvin wave response at the UTLS level
and then induce the changes in the air temperature near the tropopause (e.g., Gill,
1980; Highwood and Hoskins, 1998). The trends of the 100 hPa streamfunction show
a Rossby wave response over the TWP and a Kelvin wave response over the tropical
eastern Pacific in the Control simulation (Fig. 7d), which is caused by the changes in
the deep convection over the tropical Pacific. The Rossby-Kelvin wave response
further leads to the decrease the CPTT over the TWP and the increase of the CPTT

over the tropical eastern Pacific. Previous studies suggest that the lower stratospheric

water vapor is mainly influenced by the coldest temperature near the tropopause (e.g.,

Garfinkel et al., 2018; Zhou et al., 2021). Since the TWP has the coldest CPTT in the

boreal winter (e.g., Pan et al., 2016), the significantly decreased CPTT over the TWP

may result in significantly dried lower stratosphere (Fig. 7g). The intensity of the

upward motion over the TWP shows negative correlations with the concentration of

the tropical lower stratospheric water vapor (not shown). Hence, the enhanced upward

motion over the TWP may correspond to a dried lower stratosphere. The CPTT shows

weak trends over the TWP, and the tropical water vapor shows insignificant trends at

70 hPa in the Fixsst simulation. The comparison between the Control simulation and

the Fixsst simulation suggests that the trends of the deep convection, the CPTT, and

the lower stratospheric water vapor concentration in the tropics in NDJFM during

1958-2017 are dominated by the trends of the global SSTs, while other external

forcings may play minor roles.

Generally, the intensified upward motion may lead to more tropospheric trace

gases lifting to the upper troposphere and entering the lower stratosphere (e.g.,

Rosenlof, 2003; Lu et al., 2020). Here we use CO as a tropospheric tracer to detect the

possible influences of the enhanced upward motion over the TWP on the

transportation of the tropospheric trace gases to the upper troposphere and the lower

stratosphere. Due to the data limitation, it is not possible to show the corresponding

changes of trace gases by observations in NDJFM during 1958-2017. Here, the trends

of CO at around 200 hPa from MOPITT and MLS observations are shown in the Fig.

8. The CO increased significantly over the TWP in NDJFM in the upper troposphere
from the MOPITT (at 200 hPa during 2000-2017) and MLS data (at 215 hPa during
2005-2017). The concentration of MLS CO over the TWP is approximately 80 ppbv
at 215 hPa from MLS observations and 70 ppbv at 200 hPa from MOPITT
observations, which is consistent with previous study (e.g., Huang et al., 2016). The
MLS CO data show that the area-averaged CO increased approximately $2.0\pm3.7$ ppbv
decade$^{-1}$ over the TWP in NDJFM during 2005-2017. The area-averaged MOPITT CO
data show a stronger increase of approximately $5.0\pm3.1$ ppbv decade$^{-1}$ at 200 hPa
from 2000 to 2017 (significant at the 95% confidence level). It should be pointed out
that the linear trends of CO are calculated based on the satellite data which only cover
14 or 18 years due to the data limitation. Hence, the linear trends of CO may have
uncertainties particularly in the regions with large interannual variations. To partially
overcome this shortage, the trends of MLS CO at 215 hPa during time periods of
2005-2016, 2006-2016, 2006-2017, and 2007-2016 and the trends of MOPITT CO at
200 hPa during time periods of 2000-2016, 2001-2016, 2001-2017, and 2002-2016
are shown in Supplementary Fig. 6. It could be found that the CO in the upper
troposphere increased robustly over the TWP from both the MLS and MOPITT data.
Overall, though the observed CO only covers less than 20 years, the results from the
satellite data suggest a possible impact of the intensified upward motion over the
TWP on the trace gases in the upper troposphere.

To further illustrate the impacts of the enhanced upward motion on the trace gas

in the upper troposphere and lower stratosphere, the Control and Fixsst simulations
with WACCM4 are used. The trends of the CO concentrations from the Control and
Fixsst simulations as well as their differences are shown in Fig. 9. The tropical CO at
150 hPa shows significantly increasing trends both in the Control and the Fixsst
simulations at rates of 3.4 ppbv decade$^{-1}$ and 3.2 ppbv decade$^{-1}$, respectively, (Figs. 9a
and b). This suggests that the surface emission of the CO exerts the most important
effect on the increase of the tropical CO concentration. The differences of the CO
trends at 150 hPa between the Control simulation and the Fixsst simulation are also
displayed in Fig. 9c. Since the surface emission inventories of the two simulations are
the same, it can be inferred that the trends of the CO concentration in Fig. 9c are
mainly caused by the changes in the atmospheric circulation induced by the changes
in the global SSTs. The difference of the CO concentration at 150 hPa between the
Control simulation and the Fixsst simulation shows a significantly increasing trend at
a rate of 0.2±0.1 ppbv decade$^{-1}$ over the TWP (significant at the 95% confidence
level). At the same time, decreasing trends over the central Africa exist, which
resembles to the trend patterns of the vertical velocity in the lower TTL and the deep
convection (Figs. 5i and 7c). This indicates that the enhanced deep convection in the
TWP lead to the strengthened upward motion over the TWP, which results in an extra
6% increasing trend of CO in the upper troposphere over the TWP. It could also be
found that CO also increased in the mid latitudes of the southern hemisphere (Fig. 9c).
According to previous studies, the CO perturbation from the Indonesian fires at upper
troposphere could be transported to the tropical Indian Ocean by easterly winds and
then to the subtropics in the southern hemisphere through the southward flow during
boreal winter. The CO perturbation then spreads rapidly circling the globe following
the subtropical jet (Duncan et al., 2007). This is consistent with our results which
show intensified northerlies over the subtropical Indian Ocean (15°S-25°S,
60°E-100°E) at a rate of approximately 0.2 m s$^{-1}$ decade$^{-1}$ and strengthened westerlies
over the subtropical Indian Ocean and western Pacific (20°N-35°N, 60°E-160°E) at a
rate of approximately 0.3 m s$^{-1}$ decade$^{-1}$ (Figs. 5c and f).
The trends of the zonal mean CO concentration from model simulations are
displayed in Figs. 10a-c. The zonal mean CO shows significantly increasing trends at
all levels in the Control simulation and the Fixsst simulation, while the difference of
the zonal mean CO between the Control simulation and the Fixsst simulation shows
significantly increasing trends in the TTL but negative trends in the middle
troposphere in the tropics and the Northern Hemisphere. At the same time, the
difference of CO concentration between the Control simulation and the Fixsst
simulation averaged in the western Pacific (100°E-180°E) shows significantly
increasing trends in the tropics (20°S-10°N) from the surface to the TTL (Fig. 10f).
The CO in the layer 150-70 hPa over the TWP increased 3.2 ppbv decade$^{-1}$ and 2.8
ppbv decade$^{-1}$ in the Control and Fixsst simulations in NDJFM during 1958-2017,
respectively. And the CO difference between the Control and Fixsst simulations
increased 0.4±0.2 ppbv decade$^{-1}$ (significant at the 95% confidence level) in the layer
150-70 hPa over the TWP, which suggests that the intensifying upward motion over
the TWP and the tropical upwelling of BDC could lead to an extra 14% increasing
trend of CO. This indicates that the increased zonal mean CO in the TTL (Fig. 10c) is
mainly transported through the western Pacific bands and highlights the importance of
the upward motion over the TWP in elevating trace gases from the surface to the
upper troposphere.

To understand the CO trends in the Control and Fixsst simulations and their

differences, the trends of vertical velocity averaged over the globe and the TWP band
are given in Fig. 11. The zonal mean $w$ shows weak and even decreasing trends in the
tropics while the $w$ over the TWP intensified in the Control simulation in NDJFM
during 1958-2017. This is consistent with Fig. 5. While the SSTs fixed to
climatological values, the zonal mean $w$ shows weak trends and the $w$ over the TWP
shows significantly negative trends. The changes in the global SSTs therefore leads to
the increase of the $w$ over the TWP region as indicated in the differences between the
two simulations in Fig. 11f. In summary, the CO shows increasing trends (3.5 ppbv
decade$^{-1}$) at 150 hPa over the TWP in NDJFM during 1958-2017 induced by the
changes in the surface emissions and the upward motion. The trends of CO at 150 hPa
over the TWP in NDJFM during 1958-2017 in the Fixsst simulation mainly include
the impact induced by the increased surface emissions since the upward motion over
the TWP in the Fixsst simulation shows weak trends. The difference between the
Control and Fixsst simulations indicates that the enhanced tropospheric upward
motion over the TWP forced by the changes in the global SSTs leads to some extra
increase of CO concentrations in the upper troposphere. It should be mentioned that
the increasing trends of CO in the lower troposphere in Fig. 10f may be mainly caused
by the changes in the horizontal winds. Girach and Nair (2014) suggested that

enhanced deep convection and the subsequent intensified upward motion may lead to a decreased CO concentration in the lower troposphere and an increased CO concentration in the upper troposphere. The trends of horizontal winds at 925 hPa are shown in Supplementary Fig. 8c. There are northerly trends over east Asia and northeasterly trends near the south Asia (Supplementary Fig. 8c), which suggests that more CO-rich air from east Asia and south Asia could be transported to the TWP in the Control simulation comparing to the Fixsst simulation. Since the CO concentration in the lower troposphere over the northern Pacific is higher than that over southern Pacific, the northerly trends over the western and central Pacific may also contribute to the increased CO in the lower troposphere over the TWP in Fig. 10f.

As discussed in the Introduction, the tropospheric trace gases enter the stratosphere mainly through the large-scale tropical upwelling associated with the BD circulation. The trends of the BD circulation in different model simulations as well as their differences are displayed in Fig. 12. The tropical upwelling of BDC ($w^*$) calculated using the TEM formula increased significantly in the lower stratosphere over past decades as seen in the JRA55 data and the Control simulation (Figs. 12a and 12b). We found that the 70 hPa upward mass flux in NDJFM in the tropics (15°S-15°N) increased 2.8±1.9% decade$^{-1}$ (significant at the 95% confidence level) in the JRA55 data from 1958 to 2017 (Fig. 12a) and 4.6±4.3% decade$^{-1}$ (significant at the 95% confidence level) in the MERRA2 data from 1980 to 2017 (Supplementary Fig. 7b). From the ERA5 data, the 70 hPa upward mass flux in NDJFM increased in the north hemisphere (0-15°N) at a rate of 5.0±2.8% decade$^{-1}$ (significant at the 95%

confidence level), but decreased significantly in the south hemisphere (0-15°S) during
1958-2017 (Supplementary Fig. 7a). On average, the trend of the 70 hPa upward mass
flux in NDJFM in the tropics (15°S-15°N) is not significant in ERA5. In fact, many
previous studies have investigated the trends of BDC. For example, Abalos et al.
(2015) investigated the trends of BDC derived from JRA55, MERRA, and
ERA-Interim data during 1979-2012 and suggested that the BDC in JRA55 and
MERRA significantly strengthened throughout the layer 100-10 hPa with a rate of
2-5% decade$^{-1}$, while the BDC in ERA-Interim shows weakening trends. Diallo et al.
(2021) compared the trends of the BDC in the ERA5 and ERA-Interim during
1979-2018 and pointed out that the BDC in the ERA-Interim shows weakening trend
and the BDC in the ERA5 strengthened at a rate of 1.5% decade$^{-1}$ which is more
consistent with other studies. In the present study, we only focus on the trend of the
BDC in the wintertime (NDJFM) in the tropics (15°S-15°N) during 1958-2017, which
may lead to some differences between our result and that in the previous studies.
Overall, the trends of the tropical upwelling of BDC derived from JRA55, MERRA2
data and the Control simulation are similar to that in previous studies using both
reanalysis datasets and model results (e.g., Butchart et al., 2010; Abalos et al., 2015;
Fu et al., 2019; Rao et al., 2019; Diallo et al., 2021). However, the tropical upwelling
of the BDC decreased in ERA5 data in the tropics (15°S-15°N), which are different
from the results in JRA55 and MERRA2.
In the Fixsst simulation, the trend of $w^*$ is much weaker and not significant in
most areas. The changes in the global SSTs therefore play an important role in the
intensification of the shallow branch of the BDC as shown by the differences between
the two simulations in Fig. 12d. In summary, the tropical upwelling of the BDC is
likely strengthened as shown in JRA55 and MERRA2 reanalyses as well as model
simulations, although there are some uncertainties since the ERA5 data show a
negative trend. This may impact on the transport of the tropospheric trace gases from
the TTL to a higher altitude. The increased concentration of CO in the UTLS in Fig.
9c and 10f may be due to a combined effect of the strengthened tropical upwelling of
the BD circulation and the enhanced upward motion over the TWP. The enhancement
of upward motion over the TWP, which transported more tropospheric trace gases to
the upper troposphere, works together with the strengthened BD circulation under
global warming may lead to an increase of tropospheric trace gases over the TWP in
the lower stratosphere.
**4 Summary and Discussion**
The recent trends of the upward motion from the lower to the upper troposphere
in boreal winter over the TWP is investigated for the first time based on the JRA55,
ERA5, MERRA2 datasets and four WACCM4 simulations (Table 2). The upward
motion at 150 hPa over the TWP in NDJFM increased $8\pm3.1\%$ decade$^{-1}$ and $3.6\pm3.3\%$
decade$^{-1}$ in NDJFM from 1958 to 2017 in JRA55 and ERA5 reanalysis datasets,
respectively (Table 3). Despite the possible discontinuities between the radiosonde era
(after 1958) and the satellite era (after 1979), the upward motion at 150 hPa over the
TWP in NDJFM increased $7.5\pm7.1\%$ decade$^{-1}$ during 1980-2017 in MERRA2 data.
Such intensification of the upward motion over the TWP also exist in the middle and

lower troposphere in NDJFM in JRA55, ERA5, and MERRA2, which can be confirmed by the WACCM4 model simulations. Comparing the results between the Control and Fixsst simulations with WACCM4, it is found that the trend of the upward motion over the TWP is closely related to the changes in global SSTs, especially the SST warming over the eastern maritime continent and tropical western Pacific (see the results from the experiments R1 and R2 in Fig. 7). Warmer SSTs over the eastern maritime continent and tropical western Pacific (approximately 0.5 K) lead to a strengthened Pacific Walker circulation, enhanced deep convection and approximately 27% intensified upward motion at 150 hPa over the TWP as shown by the results from the experiments R1 and R2. The enhanced deep convection over the TWP could lead to a dryer lower stratosphere over the TWP, as the strong upward motion and the Rossby-Kelvin wave responses induce a colder tropopause over the TWP. It should be pointed out that the results in the present study are mainly based on the reanalyses data, and some uncertainties may exist. The availability of more high resolution observations in the future may enhance the quality of the reanalysis data.

Results from the Control simulation indicate that the CO concentrations increased significantly from the surface to the stratosphere over the TWP. The CO at 150 hPa increased at a rate of approximately 3.4 ppbv decade$^{-1}$ with increased surface emissions and the enhanced upward motion over the TWP. Specifically, an enhancement of tropospheric upward motion and subsequent upward transport of trace gases over the TWP lead to an extra 6% increasing trend of CO concentrations in the upper troposphere.

Furthermore, the upward mass fluxes at 70 hPa in the tropics (15°S-15°N) show

strengthening trends at rates of 2.8±1.9% decade$^{-1}$ and 4.6±4.3% decade$^{-1}$ in JRA55
data (during 1958-2017) and MERRA2 data (during 1980-2017) in NDJFM,
respectively, which is consistent with previous studies (e.g., Butchart et al., 2010; Fu
et al., 2019; Rao et al., 2019). However, such enhancement in tropical upward mass
flux at 70 hPa has large uncertainties since the ERA5 data show a negative and
insignificant trend (Supplementary Fig. 7a). The results from the Control and Fixsst
simulations indicate that the elevated CO in the upper troposphere is further uplifted
to the lower stratosphere by the intensified tropical upwelling of the BD circulation
due mainly to global SST warming and lead to an increase of CO in the lower
stratosphere. An extra 14% increasing trend of CO at the layer 150-70 hPa over the
TWP is derived from the Control and Fixsst simulations.

Tropospheric trace gases and aerosols have important impacts on the

stratospheric processes if they enter the stratosphere. For example, ozone-depleting
substances, $CH_4$ and $N_2O$ could influence on the stratospheric ozone significantly
(e.g., Shindell et al., 2013; Wang et al., 2014; WMO, 2018), which also modify the
temperature in the stratosphere significantly through their strong radiative effects.
Water vapor in the lower stratosphere, in particular, has a significant warming effect
on the surface climate (Solomon et al., 2010). Therefore, changes of trace gases in the
UTLS have important impacts on both tropospheric and stratospheric climate. Our
results suggest that the upward motion over the TWP and the vertical component of
the BDC at the lower stratosphere level have been intensified. These results suggest
that the emission from the maritime continent and surrounding areas may play a more
important role in the stratospheric processes and the global climate. In addition, more
very short lived substances emitted from the tropical ocean could be elevated to the
TTL by the enhanced convection and then transported into the stratosphere by the
large-scale uplifts and exert important effects on the stratospheric chemistry. However,
the quantitative impacts of the intensified upward motion over the TWP on
tropospheric and stratospheric trace gases and aerosols and their climate feedbacks
await further investigation using more observations and model simulations.

**Competing interests.** The authors declare that they have no conflict of interest.

**Author contributions.** KQ ran the models and wrote the first draft. WW provided
suggestions about the statistical methods and model simulations. WT designed the
study. RH, MX, and TW contributed to the manuscript writing. YP provided the data
used in the study. All authors contributed to the improvement of the results.


**Acknowledgements.** This research is supported by Strategic Priority Research
Program of Chinese Academy of Sciences (XDA17010106), the National Natural
Science Foundation of China (42075055) and the Supercomputing Center of Lanzhou
University.
The authors gratefully acknowledge the data used in the present study provided by the
corresponding scientific groups. The JRA55 data are from:
http://rda.ucar.edu/datasets/ds628.0/.

The SST data is obtained from HadISST:

https://www.metoffice.gov.uk/hadobs/hadisst/data/download.html.

The ERA5 data and ERA5.1 data are extracted from:

https://cds.climate.copernicus.eu/#!/search?text=ERA5&type=dataset.

The MERRA2 data are downloaded from:

https://search.earthdata.nasa.gov/search?q=MERRA2&fst0=Atmosphere.

The OLR data are from https://psl.noaa.gov/data/gridded/data.interp_OLR.html.

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

**Figure captions:**
**Fig. 1.** The climatological mean (averaged over 1958-2017) values of 150 hPa $w$ ($10^{-2}$
m s$^{-1}$) in Jan.-Dec. (a-l) derived from the JRA55 data.
**Fig. 2.** Trends of vertical velocity and horizontal winds at 150 hPa, 500 hPa, 700 hPa
in NDJFM derived from JRA55, ERA5, and MERRA2 data. The trends of horizontal
winds (arrows, units: $10^{-1}$ m s$^{-1}$ a$^{-1}$) and vertical velocity (shading, units: $10^{-4}$ m s$^{-1}$ a$^{-1}$)
at (a) 150 hPa; (d) 500 hPa; and (g) 700 hPa from JRA55 in NDJFM during
1958-2017. (b), (e), and (h) are the same as (a), (d), and (g) but for the results from
ERA5. (c), (f) and (i) are the same as (a), (d), and (g) except that the trends are during
1980-2017 and the wind field data are from MERRA2. The vertical velocity trends
over the dotted regions are statistically significant at the 95% confidence level. The
white areas denote missing values. The black rectangles denote the TWP region
(20°S-10°N, 100°E-180°E).
**Fig. 3.** The time series of the standardized intensity of the upward motion over the
tropical western Pacific (20°S-10°N, 100°E-180°E) at (a) 150 hPa; (b) 500 hPa; and
(c) 700 hPa extracted from JRA55 (red), ERA5 (black) and MERRA2 (blue) datasets.
The straight lines in each figure indicate the linear trends. The solid lines denote the
linear trends are significant at the 95% confidence level, while the dashed lines denote
the linear trends are significant at the 90% confidence level.
**Fig. 4.** Trends of (a) observed outgoing longwave radiation (OLR, units: W m$^{-2}$ a$^{-1}$)
provided by NOAA during 1974-2017; (b) cold-point tropopause temperature (CPTT,

units: $10^{-1}$ K $a^{-1}$) derived from JRA55 data and (c) SST (K $a^{-1}$) derived from HadISST during 1958-2017 in NDJFM. (d) The correlation coefficients between the intensity of the upward motion at 150 hPa over the TWP and SST in NDJFM during 1958-2017 with the linear trends removed. The trends and correlation coefficients over the dotted regions are statistically significant at the 95% confidence level. The black rectangles denote the TWP region (20°S-10°N, 100°E-180°E).

**Fig. 5.** Same as Fig. 2 but for the Control simulation (a, d, and g), Fixsst simulation (b, e, and h), and the difference between the simulations (c, f, and i). The vertical velocity trends over the dotted regions are statistically significant at the 95% confidence level. The black rectangles denote the TWP region (20°S-10°N, 100°E-180°E).

**Fig. 6.** The difference of vertical velocity (shading, units: $10^{-2}$ m $s^{-1}$) and horizontal winds (arrows, units: m $s^{-1}$) at (a) 150 hPa; (b) 500 hPa; and (c) 700 hPa in NDJFM between experiments R1 and R2. The differences between vertical velocity over the dotted regions are statistically significant at the 95% confidence level. The black rectangles denote the TWP region (20°S-10°N, 100°E-180°E).

**Fig. 7.** Same as Fig. 5 but for the trends of (a)-(c) OLR (W $m^{-2}$ $a^{-1}$), (d-f) CPTT (shading, units: $10^{-1}$ K $a^{-1}$) and 100 hPa streamfunction (contour, units: $10^{6}$ $m^{2}$ $s^{-1}$ $a^{-1}$), and (g-i) 70 hPa water vapor concentration (units: $10^{-2}$ ppmv $a^{-1}$). The trends in (a)-(c) and (g)-(i) over the dotted regions are statistically significant at the 95% confidence level. The CPTT trends in (d)-(f) over the dotted regions are statistically significant at the 95% confidence level. The black rectangles denote the TWP region (20°S-10°N, 100°E-180°E).

**Fig. 8.** The trends of CO derived from the MOPITT and MLS data. (a) The trends of
CO ($10^{-1}$ ppbv $a^{-1}$) at 215 hPa using MLS data in NDJFM during 2005-2017. (b) The
trends of CO ($10^{-1}$ ppbv $a^{-1}$) at 200 hPa using MOPITT data in NDJFM during
2000-2017. The trends of CO over the dotted region are statistically significant at the
90% confidence level.
**Fig. 9.** The trends of 150 hPa CO concentration ($10^{-4}$ ppmv $a^{-1}$) from (a) Control
simulation; (b) Fixsst simulation; and (c) difference between the Control simulation
and the Fixsst simulation in NDJFM during 1958-2017. The trends in (a)-(c) over the
dotted regions are statistically significant at the 95% confidence level. The black
rectangles denote the TWP region (20°S-10°N, 100°E-180°E).
**Fig. 10.** Latitude-pressure cross sections of the trends of (a)-(c) zonal mean CO
concentration ($10^{-4}$ ppmv $a^{-1}$) and (d)-(f) CO concentration ($10^{-4}$ ppmv $a^{-1}$) over the
TWP (100°E-180°E) in NDJFM during 1958-2017 in the Control simulation and
Fixsst simulation as well as their difference. (a) and (d) are the CO trends in the
Control simulation. (b) and (e) are the results in the Fixsst simulation. (c) and (f) are
the results derived from the difference between the Control and Fixsst simulations.
The trends over the dotted regions are statistically significant at the 95% confidence
level.
**Fig. 11.** Same as Fig. 10 but for the trends of tropospheric $w$ ($10^{-4}$ m $s^{-1}$ $a^{-1}$) and $v$ ($10^{-1}$
m $s^{-1}$ $a^{-1}$). The shadings denote the trends of the $w$ ($10^{-4}$ m $s^{-1}$ $a^{-1}$). The trends over the
dotted regions are statistically significant at the 90% confidence level.
**Fig. 12.** Trends of the BDC (vectors, units in the horizontal and vertical components

are $10^{-2}$ and $10^{-5}$ m s$^{-1}$ a$^{-1}$, respectively) calculated using the TEM formula from (a) JRA55; (b) Control simulation; (c) Fixsst simulation; and (d) difference between the Control simulation and the Fixsst simulation in NDJFM during 1958-2017. The shadings are the trends of the $w*$ ($10^{-5}$ m s$^{-1}$ a$^{-1}$). The trends of the vertical velocity over the dotted regions are statistically significant at the 90% confidence level.

**Tables:**

**Table 1.** Basic specifications of JRA55, ERA5, and MERRA2 used in this study.

| Name | Organization | Time period | Spatial resolution | Temporal resolution | Data assimilation |
|---|---|---|---|---|---|
| JRA55 | JMA | 1958-present | 55 km; L60 | 6-hourly | 4D-Var |
| ERA5 | ECMWF | 1950-present | 31 km; L137 | hourly | 4D-Var |
| MERRA2 | NASA | 1980-present | 0.5°×0.625°; | 3-hourly | 3D-Var |
| | GMAO | | L72 | | |



**Table 2.** Description of simulations with WACCM4.

| Experiment | Description |
| --- | --- |
| Control | Transient simulation. Observed greenhouse gases and solar irradiances. Prescribed SST forcing using observed SST. |
| Fixsst | Transient simulation. Observed greenhouse gases and solar irradiances. Prescribed SST forcing using monthly mean climatology from 1958 to 2017. |
| R1 | Time-slice simulation. SSTs prescribed as the climatological mean of 1998-2017 over the region 20°S-20°N, 120°E-160°E in NDJFM, but fixed as climatological mean of 1958-2017 over other regions. |
| R2 | Same as R1, but the SSTs over the region 20°S-20°N, 120°E-160°E are prescribed as the climatological mean SSTs during 1958-1977. |


**Table 3.** The trends of the upward motion over the TWP at 150 hPa, 500 hPa, and 700
hPa in NDJFM during 1958-2017 from JRA55, ERA5, MERRA2, Control simulation
and Fixsst simulation. And the trends of 150 hPa CO from the Control and Fixsst
simulations.

| | JRA55 | ERA5 | MERRA2 | Control | Fixsst |
|---|---|---|---|---|---|
| 150 hPa Upward motion | $3.0\pm1.2\times10^9$ kg s$^{-1}$ decade$^{-1}$ | $1.3\pm1.2\times10^9$ kg s$^{-1}$ decade$^{-1}$ | $3.0\pm2.8\times10^9$ kg s$^{-1}$ decade$^{-1}$ | $2.0\pm1.2\times10^9$ kg s$^{-1}$ decade$^{-1}$ | $-4.8\pm6.4\times10^8$ kg s$^{-1}$ decade$^{-1}$ |
| 500 hPa Upward motion | $4.6\pm2.6\times10^9$ kg s$^{-1}$ decade$^{-1}$ | $2.5\pm2.5\times10^9$ kg s$^{-1}$ decade$^{-1}$ | $5.4\pm5.3\times10^9$ kg s$^{-1}$ decade$^{-1}$ | $3.5\pm2.4\times10^9$ kg s$^{-1}$ decade$^{-1}$ | $-1.0\pm1.3\times10^9$ kg s$^{-1}$ decade$^{-1}$ |
| 700 hPa Upward motion | $2.9\pm1.7\times10^9$ kg s$^{-1}$ decade$^{-1}$ | $1.9\pm1.6\times10^9$ kg s$^{-1}$ decade$^{-1}$ | $3.9\pm3.8\times10^9$ kg s$^{-1}$ decade$^{-1}$ | $1.8\pm1.4\times10^9$ kg s$^{-1}$ decade$^{-1}$ | $-6.3\pm8.1\times10^8$ kg s$^{-1}$ decade$^{-1}$ |
| 150 hPa CO | -- | -- | -- | 3.4 ppbv decade$^{-1}$ | 3.2 ppbv decade$^{-1}$ |


**Figures**

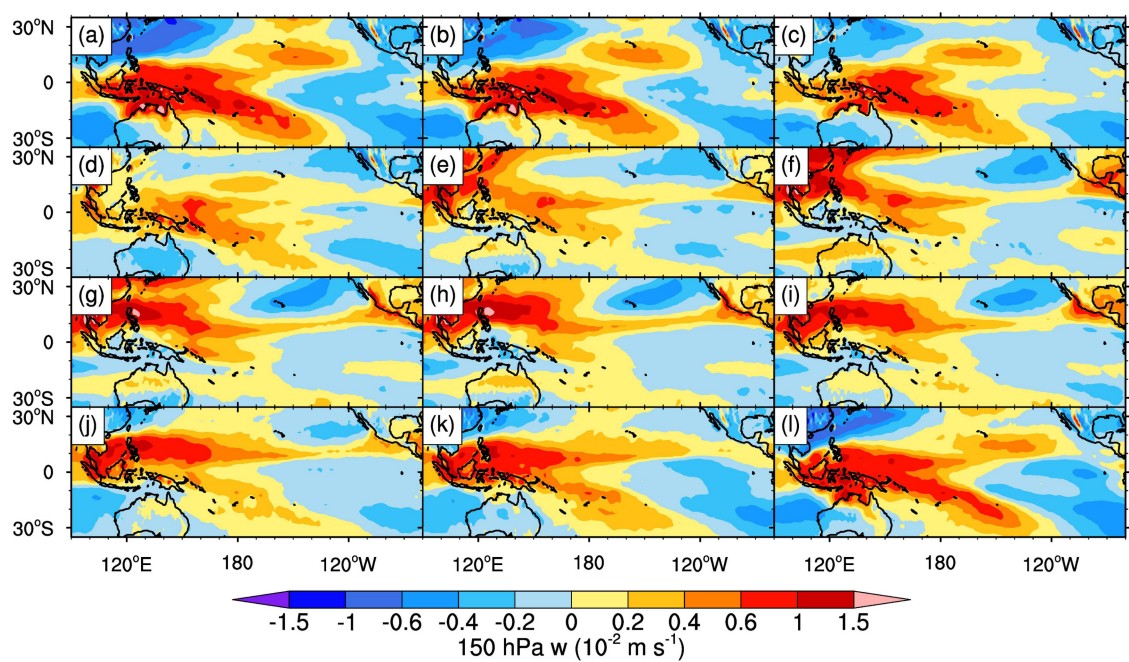

 **Fig. 1.** The climatological mean (averaged over 1958-2017) values of 150 hPa $w$ ($10^{-2}$

 m s$^{-1}$) in Jan.-Dec. (a-l) derived from the JRA55 reanalysis data.

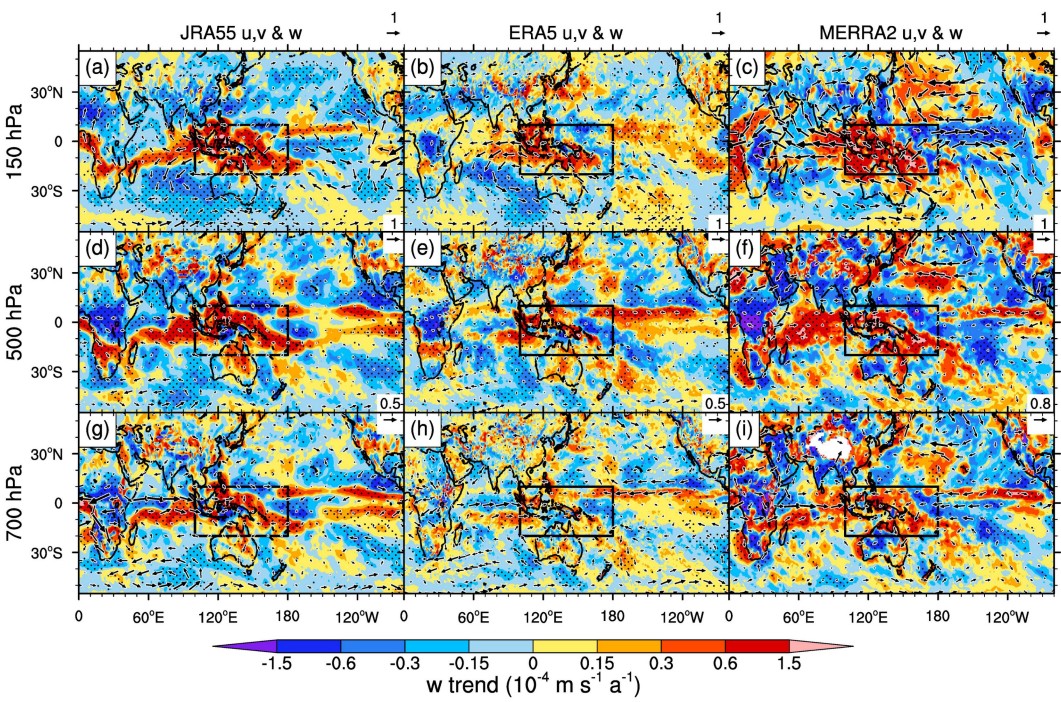

**Fig. 2.** Trends of vertical velocity and horizontal winds at 150 hPa, 500 hPa, 700 hPa in NDJFM derived from JRA55, ERA5, and MERRA2 data. The trends of horizontal winds (arrows, units: $10^{-1}$ m s$^{-1}$ a$^{-1}$) and vertical velocity (shading, units: $10^{-4}$ m s$^{-1}$ a$^{-1}$) at (a) 150 hPa; (d) 500 hPa; and (g) 700 hPa from JRA55 in NDJFM during 1958-2017. (b), (e), and (h) are the same as (a), (d), and (g) but for the results from ERA5. (c), (f) and (i) are the same as (a), (d), and (g) except that the trends are during 1980-2017 and the wind field data are from MERRA2. The vertical velocity trends over the dotted regions are statistically significant at the 95% confidence level. The white areas denote missing values. The black rectangles denote the TWP region (20°S-10°N, 100°E-180°E).


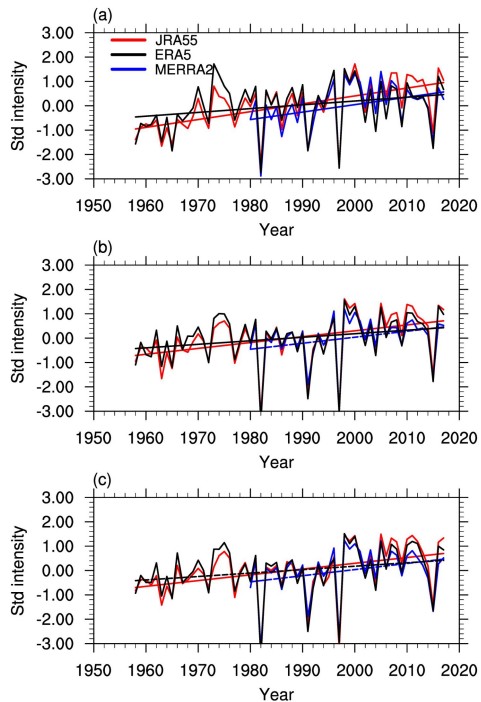


**Fig. 3.** The time series of the standardized intensity of the upward motion over the
tropical western Pacific (20°S-10°N, 100°E-180°E) at (a) 150 hPa; (b) 500 hPa; and
(c) 700 hPa extracted from JRA55 (red), ERA5 (black) and MERRA2 (blue) datasets.
The straight lines in each figure indicate the linear trends. The solid lines denote the
linear trends are significant at the 95% confidence level, while the dashed lines denote
the linear trends are significant at the 90% confidence level.

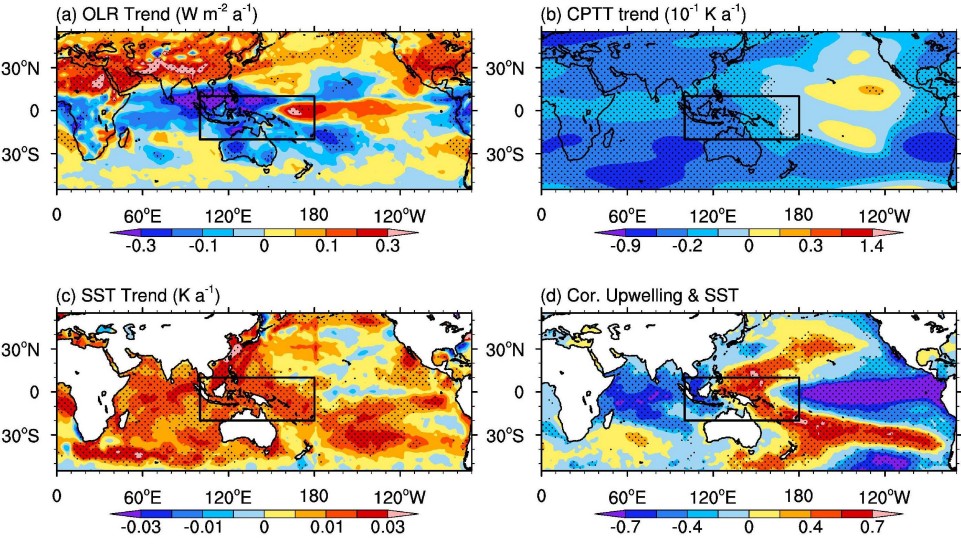


**Fig. 4.** Trends of (a) observed outgoing longwave radiation (OLR, units: W m$^{-2}$ a$^{-1}$)
provided by NOAA during 1974-2017; (b) cold-point tropopause temperature (CPTT,
units: 10$^{-1}$ K a$^{-1}$) derived from JRA55 data and (c) SST (K a$^{-1}$) derived from HadISST
during 1958-2017 in NDJFM. (d) The correlation coefficients between the intensity of
the upward motion at 150 hPa over the TWP and SST in NDJFM during 1958-2017
with the linear trends removed. The trends and correlation coefficients over the dotted
regions are statistically significant at the 95% confidence level. The black rectangles
denote the TWP region (20°S-10°N, 100°E-180°E).

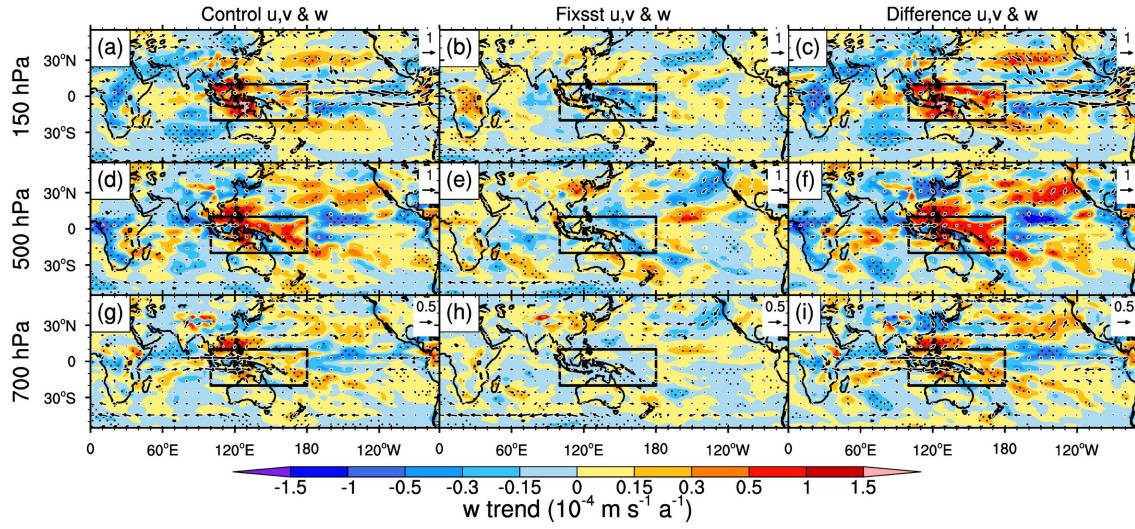


**Fig. 5.** Same as Fig. 2 but for the Control simulation (a, d, and g), Fixsst simulation (b,

e, and h), and the difference between the simulations (c, f, and i). The vertical velocity

trends over the dotted regions are statistically significant at the 95% confidence level.

The black rectangles denote the TWP region (20°S-10°N, 100°E-180°E).



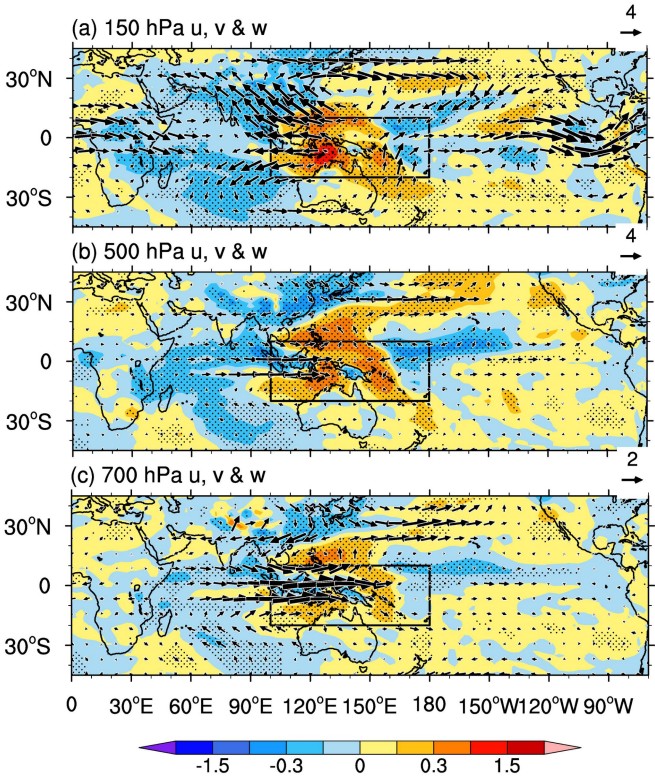


**Fig. 6.** The difference of vertical velocity (shading, units: $10^{-2}$ m s$^{-1}$) and horizontal
winds (arrows, units: m s$^{-1}$) at (a) 150 hPa; (b) 500 hPa; and (c) 700 hPa in NDJFM
between experiments R1 and R2. The differences between vertical velocity over the
dotted regions are statistically significant at the 95% confidence level. The black
rectangles denote the TWP region (20°S-10°N, 100°E-180°E).

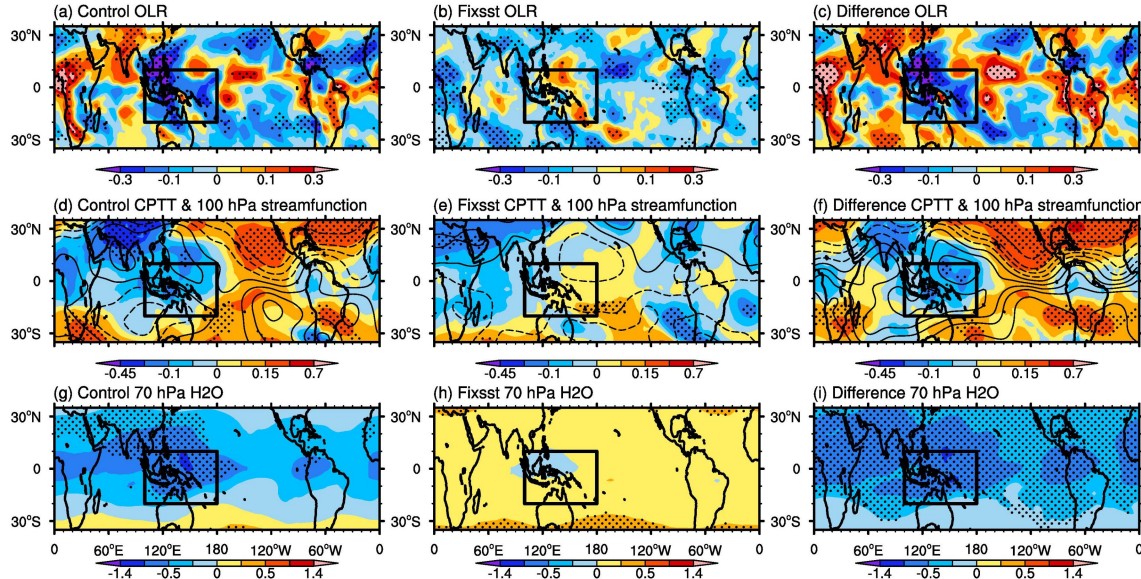


**Fig. 7.** Same as Fig. 5 but for the trends of (a)-(c) OLR (W m$^{-2}$ a$^{-1}$), (d-f) CPTT (shading, units: $10^{-1}$ K a$^{-1}$) and 100 hPa streamfunction (contour, units: $10^6$ m$^2$ s$^{-1}$ a$^{-1}$), and (g-i) 70 hPa water vapor concentration (units: $10^{-2}$ ppmv a$^{-1}$). The trends in (a)-(c) and (g)-(i) over the dotted regions are statistically significant at the 95% confidence level. The CPTT trends in (d)-(f) over the dotted regions are statistically significant at the 95% confidence level. The black rectangles denote the TWP region (20°S-10°N, 100°E-180°E).

1170

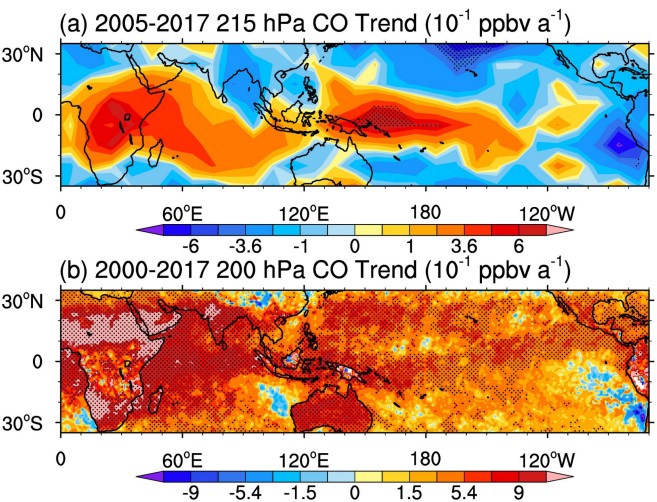

1171

Fig. 8. The trends of CO derived from the MOPITT and MLS data. (a) The trends of

CO ($10^{-1}$ ppbv a$^{-1}$) at 215 hPa using MLS data in NDJFM during 2005-2017. (b) The

trends of CO ($10^{-1}$ ppbv a$^{-1}$) at 200 hPa using MOPITT data in NDJFM during

2000-2017. The trends of CO over the dotted region are statistically significant at the

90% confidence level.



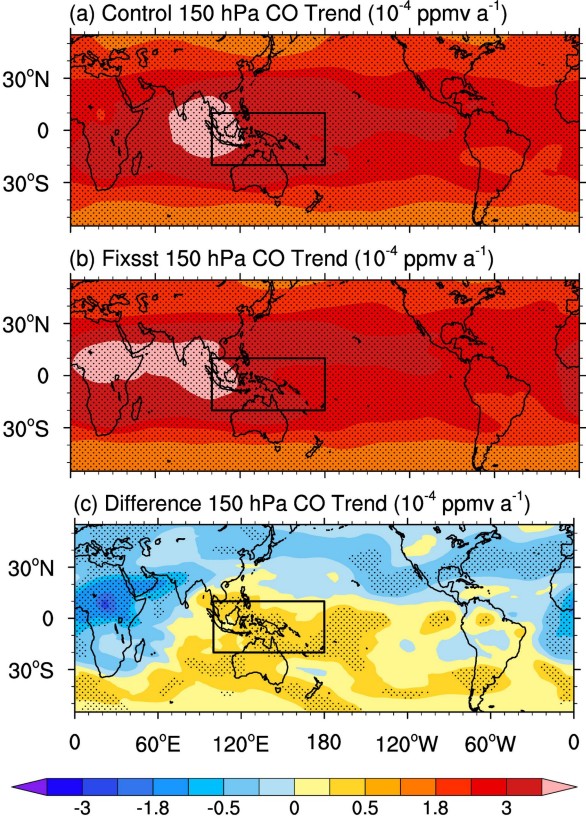


**Fig. 9.** The trends of 150 hPa CO concentration ($10^{-4}$ ppmv $a^{-1}$) from (a) Control
simulation; (b) Fixsst simulation; and (c) difference between the Control simulation
and the Fixsst simulation in NDJFM during 1958-2017. The trends in (a)-(c) over the
dotted regions are statistically significant at the 95% confidence level. The black
rectangles denote the TWP region (20°S-10°N, 100°E-180°E).



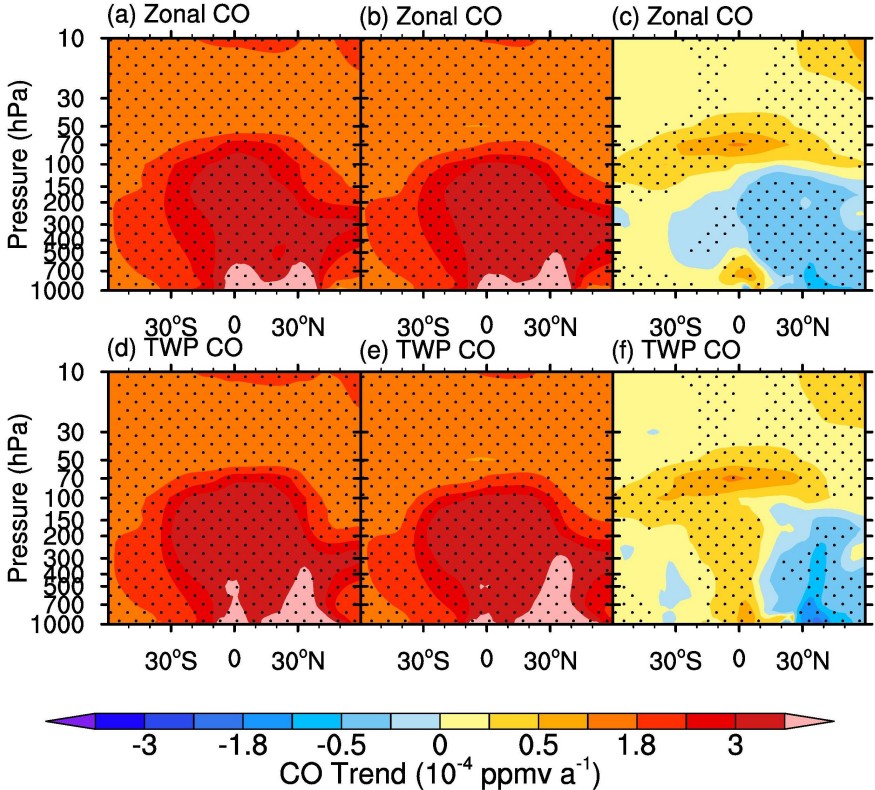


**Fig. 10.** Latitude-pressure cross sections of the trends of (a)-(c) zonal mean CO

concentration ($10^{-4}$ ppmv a$^{-1}$) and (d)-(f) CO concentration ($10^{-4}$ ppmv a$^{-1}$) over the

TWP (100°E-180°E) in NDJFM during 1958-2017 in the Control simulation and

Fixsst simulation as well as their difference. (a) and (d) are the CO trends in the

Control simulation. (b) and (e) are the results in the Fixsst simulation. (c) and (f) are

the results derived from the difference between the Control and Fixsst simulations.

The trends over the dotted regions are statistically significant at the 95% confidence

level.



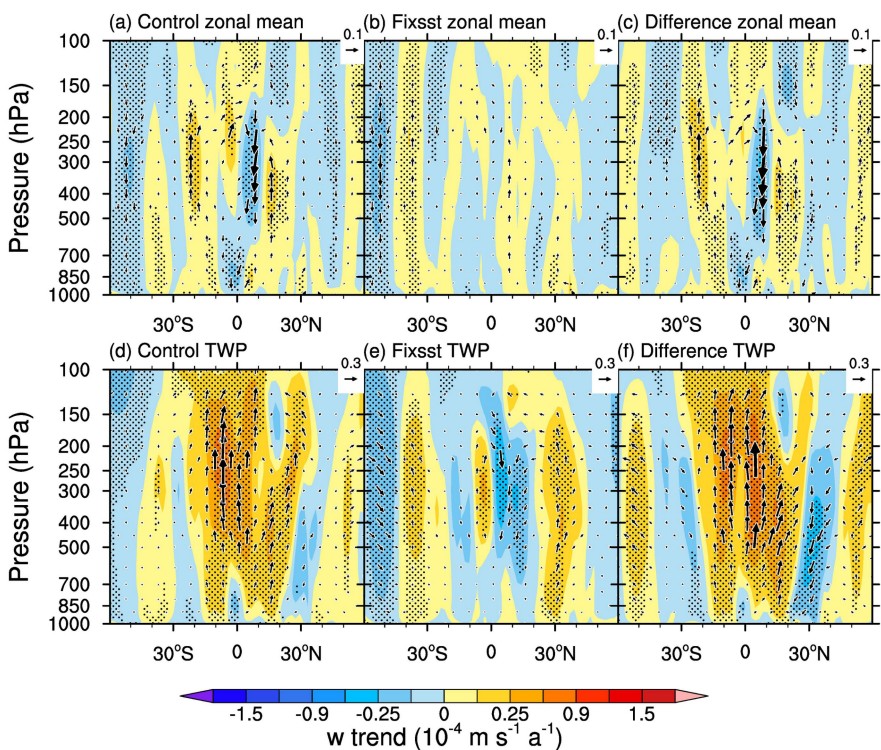


**Fig. 11.** Same as Fig. 10 but for the trends of tropospheric $w$ ($10^{-4}$ m s$^{-1}$ a$^{-1}$) and $v$ ($10^{-1}$
m s$^{-1}$ a$^{-1}$). The shadings denote the trends of the $w$ ($10^{-4}$ m s$^{-1}$ a$^{-1}$). The trends over the
dotted regions are statistically significant at the 90% confidence level.

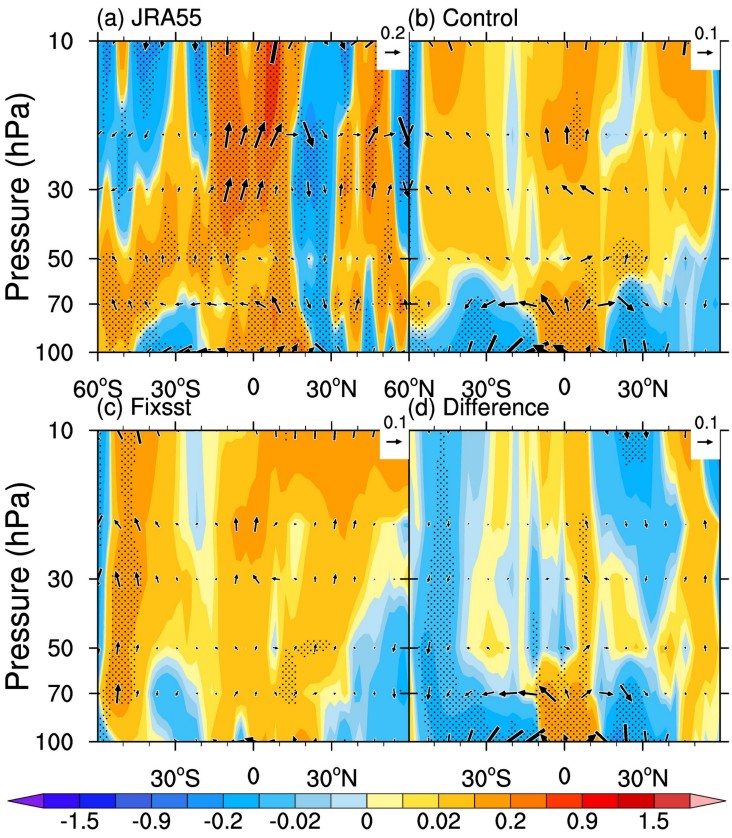

**Fig. 12.** Trends of the BDC (vectors, units in the horizontal and vertical components are $10^{-2}$ and $10^{-5}$ m s$^{-1}$ a$^{-1}$, respectively) calculated using the TEM formula from (a) JRA55; (b) Control simulation; (c) Fixsst simulation; and (d) difference between the Control simulation and the Fixsst simulation in NDJFM during 1958-2017. The shadings are the trends of the $w*$ ($10^{-5}$ m s$^{-1}$ a$^{-1}$). The trends of the vertical velocity over the dotted regions are statistically significant at the 90% confidence level.