# Peer review of "Enhanced upward motion through the troposphere over the tropical western Pacific and its implications for the transport of trace gases from the troposphere to the stratosphere"

_Atmospheric Chemistry and Physics, 2021_

## Author Comment (AC1)

**Responses to the comments by Referee #1**

**Manuscript number**: acp-2021-647

**Title**: **Enhanced upward motion through the troposphere over the tropical**

**western Pacific and its implications for the transport of trace gases from the**

**troposphere to the stratosphere**

**Author(s)**: Kai Qie, Wuke Wang, Wenshou Tian[*], Rui Huang, Mian Xu, Tao Wang,

Yifeng Peng

**December 2021**

This is an interesting and useful study. However the scientific content, the quality of the study and its presentation should be improved. In particular, the text is in some parts very descriptive and technical. I suggest some major revisions before publication by ACP.

**Re: Thank you very much for your helpful suggestions which help us improve our manuscript substantially. We have modified our manuscript according to the comments. Our point-to-point responses to the reviewer' s comments are below:**

**General comments:**

1) In general in the manuscript it is very often written 'we found a positive or negative trend'. Please specify here your message by adding some numbers in the text (a trend of xxx per year or a change of xxx within 60 years from 1958 to 2017). It would be also very helpful to give the reader an impression whether these trends are of minor or major importance by adding some numbers from the literature for comparison. In general, I am wondering that the results are not discussed more quantitatively (see specific comments below). Further, please explain in detail how the trends are calculated and how the El Niño Southern Oscillation (ENSO) is considered in calculating the trends.

**Re: We thank the reviewer for the constructive comments. The quantitative results are added to the revised manuscript according to the referee's specific comments below. The methods of how the trends are calculated and how the impact of ENSO is evaluated are also described in the revised manuscript. The details are shown in the responses to the referee's specific comments below.**

2) Figures: In general, the font size of the labels is very small and should be enlarged. Further, the text in the figure captions is very similar to each other. Please give here the reader more information which data or model simulations are shown and add some explanation what is important or what is the main message of the figure.

**Re: Thanks for the suggestion. The font sizes of the labels in each figure are**
**enlarged, and the figure captions are rephrased.**

3) In Section 2 the used data sets and model simulations are described. However, I am
missing a bit more motivation for the reader to understand why these data sets and
model simulations are used.   A bit more explanation would be helpful.

**Re: Thanks for the comment. We have added some text to explain why the**
**datasets and model simulations are used in this study, and the descriptions about**
**the reanalysis datasets and model simulations are rephrased according to the**
**referee's specific comments.**

3) The use of observations such as CO satellite measurements would strengthen the
main message of the manuscript. Therefore, I recommend to add some satellite data
(e.g. MLS CO https://mls.jpl.nasa.gov/eos-aura-mls/data-products/co)

**Re: We thank the reviewer's good suggestion. An extra figure showing the trends**
**of CO observed by MOPITT and MLS at near 200 hPa during 2000-2017 and**
**2005-2017 is added in the revised manuscript. The CO shows significantly**
**increasing trends over the TWP in NDJFM using MOPITT (at 200 hPa during**
**2000-2017) and MLS data (at 215 hPa during 2005-2017). The MLS CO data**
**show that the area-averaged CO increased approximately 2.0±3.7 ppbv decade$^{-1}$**
**over the TWP, while the CO increased 5.0±3.1 ppbv decade$^{-1}$ near the equator,**
**150°E at 215 hPa in NDJFM during 2005-2017 (Fig. R1). The area-averaged**
**MOPITT CO data increased at a rate of 5.0±3.1 ppbv decade$^{-1}$ at 200 hPa over**
**the TWP in NDJFM during 2000-2017. It should be pointed out that the linear**
**trends of CO are calculated based on the satellite data which only cover 14 or 18**
**years due to the data limitation here. Hence, the linear trends of CO may have**
**uncertainties particularly in the regions with large interannual variations in CO.**
**To partially overcome this shortage, the trends of MLS CO at 215 hPa during**

time periods of 2005-2016, 2006-2016, 2006-2017, and 2007-2016 and the trends of MOPITT CO at 200 hPa during time periods of 2000-2016, 2001-2016,

2001-2017, and 2002-2016 are shown in Fig. R2 (Supplementary Fig. 6). It could be found that the CO near 200 hPa shows robustly increasing trends over the

TWP in satellite data (both of MLS and MOPITT). Overall, though the observed

CO only covers less than 20 years, the results from the satellite data may provide extra evidence for the impact of the positive trends of upward motion over the

TWP on the trace gases in the upper troposphere. The above discussion is added to the revised manuscript. We hope these results may further support our main conclusions in this study.

[Figure]

Fig. R1. The trends of CO derived from the MLS and MOPITT data. (a) The trends of CO ($10^{-1}$ ppbv $a^{-1}$) at 215 hPa using MLS data in NDJFM during

2005-2017. (b) The trends of CO ($10^{-1}$ ppbv $a^{-1}$) at 200 hPa using MOPITT data in NDJFM during 2000-2017. The trends of CO over the dotted region are statistically significant at the 90% confidence level.

[Figure]

**Fig. R2. The trends of CO derived from the MLS and MOPITT data. (a)-(d) The**
**trends of CO ($10^{-1}$ ppbv $a^{-1}$) at 215 hPa using MLS data in NDJFM during**
**periods of (a) 2005-2016; (b) 2006-2016; (c) 2006-2017; and (d) 2007-2016. (e)-(h)**
**The trends of CO ($10^{-1}$ ppbv $a^{-1}$) at 200 hPa using MOPITT data in NDJFM**
**during periods of (e) 2000-2016; (f) 2001-2016; (g) 2001-2017; and (h) 2002-2016.**
**The trends of CO over the dotted region are statistically significant at the 90%**
**confidence level.**

**Specific Comments:**

P2 L2: 'A significantly intensified upward motion through the troposphere over the
TWP in the boreal wintertime (November to March of the next year) has been
detected.' Please make this statement more quantitative.

**Re: Corrected. The phrase is rewritten as: "A significantly intensified upward**
**motion through the troposphere over the TWP in the boreal wintertime**
**(November to March of the next year, NDJFM) has been detected using multiple**
**reanalysis datasets. The upward motion over the TWP is intensified at rates of**
**8±3.1% decade$^{-1}$ and 3.6±3.3% decade$^{-1}$ in NDJFM at 150 hPa from 1958 to 2017**

**using JRA55 and ERA5 reanalysis datasets, while the MERRA2 reanalysis data**

**show a 7.5±7.1% decade$^{-1}$ intensified upward motion for the period 1980-2017."**

P2 L18: Please specify here which reanalyses are used.

**Re: Added.**

P2 L23: 'numerical simulation' -->   'simulation with WACCM4' ?

**Re: Updated.**

P2 L24: 'show that more CO could be elevated to the tropical tropopause layer (TTL)'

Please make this statement more quantitative.

**Re: Rephrased as: "Using CO as a tropospheric tracer, the WACCM4**

**simulations show that an increase of CO at a rate of 0.4 ppbv decade$^{-1}$ at the**

**layer 150-70 hPa in the tropics is mainly resulted from the global SST warming**

**and the subsequent enhanced upward motion over the TWP in the troposphere**

**and strengthened tropical upwelling of Brewer-Dobson (BD) circulation in the**

**lower stratosphere."**

P2 L27: Why is aerosol explicitly emphasized here. Please clarify (e.g. outflow from polluted air from South Asia?)

**Re: We thank the reviewer's comment. This sentence has been rewritten as:**

**"This implies that more tropospheric trace gases and aerosols from both**

**natural maritime source and outflow from polluted air from South Asia may**

**enter  the stratosphere through the TWP region and affect the stratospheric**

**chemistry and climate."**

P3  L42: Please add possible sources of ozone-depleting halogen-containing substances in TWP (outflow from anthropogenic emissions from South Asia, natural maritime bromine-containing substances?).

**Re: We thank the reviewer's comment. This sentence has been rewritten as:**
**"Through the TWP region, tropospheric trace gases, e.g., the natural maritime**
**bromine-containing substances and outflow from anthropogenic emissions from**
**South Asia, are lifted to the upper troposphere by the strong upward motion and**
**the deep convection and subsequently into the stratosphere by the large-scale**
**upwelling (e.g., Levine et al., 2007, 2008; Navarro et al., 2015), which affect the**
**ozone concentration and other chemical processes in the stratosphere (e.g., Feng**
**et al., 2007; Sinnhuber et al., 2009)."**

P4 L45: (Saiz-Lopez and von Glasow, 2012; Wang et al., 2015). -> (e.g.
Saiz-Lopez ...).

**Re: Corrected.**

P4 L46: 'the coldest tropopause' of what? Please specify.

**Re: Here we mean that the TWP region has the lowest tropopause temperature**
**over the globe. Corrected as "At the same time, the TWP region has the lowest**
**cold-point tropopause temperature (CPTT) over the globe and plays an**
**important role in controlling the water vapor concentration in the stratosphere."**

P4 L49: 'an important region for troposphere-to-stratosphere transport' Please add
some references.

**Re: Added.**

P4 L50: Is the TWP more important for stratospheric chemistry as other regions in the
atmosphere? Please clarify?

**Re: We thank for the reviewer's comment. Here we want to summarize the**
**importance of the TWP region. The sentence was modified as "The TWP is an**
**important region for tropospheric trace gases being transported from the**
**troposphere to the stratosphere, and therefore influencing the stratospheric**

**chemistry (e.g., Fueglistaler et al., 2004; Levine et al., 2007; Krüger et al., 2008;**
**Pan et al., 2016) ."**

P4 L66-70: The impact of ozone-depleting halogen-containing substances is already
mentioned on P3 L42. I propose to combine these two sentences in one paragraph.

**Re: These sentences are combined in the first paragraph of Introduction section**
**as: "Through the TWP region, tropospheric trace gases, e.g., the natural**
**maritime bromine-containing substances and outflow from anthropogenic**
**emissions from South Asia, are lifted to the upper troposphere by the strong**
**upward motion and the deep convection and subsequently into the stratosphere**
**by the large-scale upwelling (e.g., Levine et al., 2007, 2008; Navarro et al., 2015),**
**which affects the ozone concentration and other chemical processes in the**
**stratosphere (e.g., Feng et al., 2007; Sinnhuber et al., 2009)."**

P4 L71: 'Based on a trajectory model, Fueglistaler et al. (2004) pointed out that the
TWP region is a primary source of the tropospheric air entering the stratosphere and
approximately 80% of the trajectories ascending into the stratosphere enter the TTL
from the TWP'.   However, in L63 it is written: 'the TWP is not the dominant entry of
trace gases transported from the troposphere into the lower stratosphere'. Please
rephrase this statement more carefully.

**Re: Thanks for the comment. The statement is rephrased as: "Though the**
**vertical transport from TTL to the lower stratosphere is dominated by the BD**
**circulation, numerous studies confirmed that the TWP region is an important**
**pathway of the surface air entering the TTL (Fueglistaler et al., 2004; Levine et**
**al., 2007; Krüger et al., 2008; Haines and Esler, 2014). Based on a trajectory**
**model, Fueglistaler et al. (2004) pointed out that approximately 80% of the**
**trajectories ascending into the stratosphere from the TTL are originated from**
**the TWP region."**

P6 L100: 'using reanalysis datasets and model simulations' --> 'using JRA55, ERA5 and MERRA2 reanalysis and different WACCAM4 simulations as described in Sect. 2.'

**Re: Corrected.**

P6 L102: 'is also discussed.' --> ' will be discussed in Sect. 3'

**Re: Corrected.**

P6 L110: Please add the horizontal resolution of ERA5 data (0.3° × 0.3°), which is much higher as in JRA55 and MERRA2. What about differences in vertical and temporal resolution. Please specify.

**Re: Thanks for the comment. The description of the JRA55, ERA5 and MERRA2 datasets are rephrased in Section 2, and the information about the vertical, horizontal, and temporal resolution are added.**

P6 L124: 'UTLS' is not yet introduced in the text.

**Re: Corrected.**

P6 L125: 'even though there are still large biases in the reanalysis datasets' What are the differences between the three different reanalyses (JRA55, ERA5 and MERRA2) used here? Please specify.

**Re: According to the results of Uma et al. (2021), the description is added to the manuscript as: "the updrafts from the JRA55 data in the UTLS are stronger than those from ERA5 and MERRA2 data." It should be mentioned that Uma et al. (2021) did not give quantitative differences between them.**

P8 L145: 'except that the global SSTs are fixed to the climatological mean values during 1955-2018 (long-term mean for each calendar month during 1955-2018.' Why are the SST not fixed to a value representative for the beginning of the 60-year period?

**Re: The Control and Fixsst simulations are designed to investigate the impact of**

**SST changes on the intensified upward motion over the TWP. For this purpose,**

**using the SST climatology representative for the beginning of the 60-year period**

**to force the simulation should also be proper. Since we compare the trends**

**between the Control (transient) and the Fixsst (constant) simulations, the state of**

**the Fixsst simulation should not influence the results. The SSTs are fixed to the**

**mean of 1958-2017 rather than 1960s to make the mean state of the two**

**simulations more consistent with each other.**

P8 L146 Please explain the added-value of a time-slice experiment compared to the hindcast simulation.

**Re: Thanks for the comment. The SSTs in the hindcast simulation are prescribed**

**as the observed SSTs, with changes of SSTs over the globe. SSTs in the time-slice**

**simulations are only modified in the eastern maritime continent and the tropical**

**western Pacific (20°S-20°N, 120°E-160°E) , which emphasizes the importance of**

**the SSTs over these areas. The descriptions are clarified in the revised**

**manuscript.**

P8 L150: For better motivation, please explain in more detail why this set up is used for the two time-slice simulations.

**Re: Thanks for the suggestion. Some explanations are added to the manuscript**

**as: "To figure out the impact of the warming SST over the TWP region on the**

**intensifying trend of the upward motion over the TWP region, a couple of**

**time-slice simulations (R1 and R2) are also integrated for 33 years… Since the**

**SSTs over the TWP show significantly warming trends, the SSTs during 1998-2017 are higher than the SSTs during 1958-1977. Hence, the difference between R1 and R2 reflect the impact of the warmed SSTs over the TWP on the atmospheric circulation."**

P9 L171: 'the climatological distribution of the vertical velocity at 150 hPa for each month of the year.' --> Mean values of the vertical velocity at 150 hPa for each month averaged over 60 years from 1958 to 2017. Yes?

**Re: Yes. The statement is corrected correspondingly.**

Why is JRA55 and not ERA5 or MERRA2 selceted for Fig.1? What are the difference between JRA55 and ERA5/MERRA2?

**Re: The pattern of the 150 hPa vertical velocity from JRA55 data shown in Fig. 1 is similar to the patterns of the 150 hPa vertical velocity from ERA5 and MERRA2 datasets. To avoid repetition, only the result from JRA55 data is shown in Fig. 1. According to the referee's comment, the climatological mean vertical velocity in NDJFM in ERA5 and MERRA2 is added to the supplementary material. The vertical velocity differences between JRA55 and the ERA5 and MEERA2 data are further discussed in the revised manuscript.**

P9 L180: please add text within ++: 'which is more important to the transport of air over the TWP from the lower troposphere to the TTL +compared to the summer months (as shown in Fig. 1) + and subsequently to the lower stratosphere.

**Re: Corrected.**

P9 L182: 'Notably, the 150 hPa w shows no subsidence over the maritime continent, while there is descending motion over the maritime continent at 100 hPa (not shown), which is referred to the "stratospheric drain" (Gettleman et al., 2000; Sherwood, 2000).' The 100 hPa values should be shown in an electronic supplement.

**Re: The 100 hPa *w* values using JRA55, ERA5 and MERRA2 are shown in**

**Supplementary Fig. 2.**

P10 L186: Please explain in detail how the trend is calculated.

**Re: We thank for the reviewer's suggestion. The description about the trend and**

**the significance test is added to Section 2 as:**

**"Linear trends and the significance test. The linear trends are estimated**

**using a simple least square regression method. The significances of the**

**correlation coefficients, mean differences, and trends are determined via a**

**two-tail Student's t-test. The confidence interval of trend is calculated using the**

**following equation (Shirley et al., 2004):** $\left( b - t_{1-\frac{\alpha}{2}}(n-2)\sigma_b, b + t_{1-\frac{\alpha}{2}}(n-2)\sigma \right)$

**where b is the estimated slope,** $\sigma$ **denotes the standard error of the slope, and**

$t_{1-\frac{\alpha}{2}}(n-2)$ **represents the value of t-distribution with the degree of freedom**

**equal to *n*-2. α is the two-tailed confidence level.** $\sigma$ **is calculated as:**

$\sigma = b\sqrt{\dfrac{\dfrac{1}{r^2}-1}{n-2}}$ **."**

P10 L187: 'using reanalysis datasets' -> 'using JARA55, ERA5 and MERRA2

reanalyses.'

**Re: Corrected.**

P10 L191: ->'is intensifying through the troposphere from 1958 to 2017.'

**Re: Corrected.**

P10 L193 : add 'used here' or 'used in this study'

**Re: Added.**

Figure 2: In MERRA2 the horizontal winds seems to be much stronger compared to

JARA55 and ERA5. Could you make a comment on this. Please discuss the similarities and differences of the three reanalyses in more detail. Maybe you could show an additional figure showing the differences of ERA5 and MERRA2 compared to JARA55. ERA5 has much higher spacial and temporal resolution as JRA55 and

MERRA2, therefore I would expect pronounced differences to JARA55 and

MERRA2, in particular convection is much improved compared to the previous

ECMWF reanalysis ERA-Interim.

**Re: Thanks for the comment. In Fig. 2, the trends of the horizontal winds seem**

**to be much stronger in MERRA2 compared to JRA55 and ERA5. It should be**

**noted that the wind trends in JRA55 and ERA5 are calculated during the period**

**1958-2017, however, the wind trends of in MERRA2 are calculated during the**

**period 1980-2017. To further figure out whether there are large differences**

**between the trends of the winds between JRA55, ERA5 and MERRA2, the**

**trends of winds during 1980-2017 in NDJFM derived from JRA55, ERA5 and**

**MERRA2 are shown here (and in the supplementary material). It could be seen**

**that the trends of horizontal winds in Figs. R3a and R3b are larger than the**

**trends of horizontal winds in Figs. 2a and 2b (in manuscript). And there are**

**insignificant differences between the trends of horizontal winds in JRA55, ERA5,**

**and MERRA2. Hence, the differences of the trends of the horizontal winds in Fig.**

**2 are mainly due to the different time periods which are used to calculate the**

**trends. The trend patterns of the winds in JRA55, ERA5, and MERRA2 are**

**similar. However, there are also some differences between the trends of vertical**

**velocity in JRA55, ERA5, and MERRA2. There are significantly positive trends**

**over the TWP regions in JRA55, ERA5, and MERRA2, while the positive trends**

**of vertical velocity over the TWP in ERA5 seem to be weaker than those in**

**JRA55 and MERRA2. Comparing to the negative trends of the vertical velocity**

**over the central Pacific in JRA55 and ERA5, the negative trends of the vertical**

**velocity over the central Pacific in MERRA2 extend more northward. The above**

**discussion is added to the corresponding paragraph in the revised manuscript.**

[Figure]

**Fig. R3. The trends of the vertical velocity and horizontal winds in NDJFM using**

**JRA55 (a, d, g), ERA5(b, e, h) and MERRA2(c, f, i) data during 1980-2017 at**

**different levels. (a)-(c) are the trends of winds at 150 hPa. (d)-(f) are the trends of**

**winds at 500 hPa. (g)-(i) are the trends of winds at 700 hPa. The trends of**

**vertical velocity over the dotted region are statistically significant at the 90%**

**confidence level.**

Figure 3: Please Explain how 'standardized intensity' is calculated. What is the reason for the extreme minima (1981, 1991, 1999)?   El Niño Southern Oscillation (ENSO)?

**Re: "The intensity of the upward motion over the TWP is simply defined as the**

**area-averaged upward mass flux at a specific level. And the standardized**

**intensity is the intensity divided by the standard deviation of the intensity at the**

**corresponding level." The explanation of the standardized intensity is added to**

the manuscript. The extreme minima (actually, the years are 1982, 1991, and

1997) are mainly due to the ENSO events (El Niño), which may result in a weak upward motion over the TWP (e.g., Levine et al., 2008; Hosking et al., 2012; Hu et al., 2016). To figure out the influence of the El Niño events (1982, 1991, 1997), the time series of the standardized intensity of the upward motion over the TWP

in NDJFM after removing the ENSO signal using the linear regression method (Hu et al., 2018) in JRA55, ERA5, and MERRA2 are shown here (Fig. R4 and

Supplementary Fig. 5). It could be seen that the extreme minima become much weaker after removing the ENSO signal using the linear regression method. This result suggests that the El Niño events could affect the upward motion over the

TWP and to a large extent result in the extreme minima (1982, 1991, and 1997).

Notably, the upward motions over the TWP at 150 hPa, 500 hPa, and 700 hPa in

NDJFM in JRA55, ERA5, and MERRA2 still show statistically significant intensifying trends after removing the ENSO signal in Supplementary Fig. 5, which suggests that ENSO events exert limited impacts on the trends of the upward motion over the TWP in NDJFM during 1958-2017. Some of above discussions are added to the revised manuscript.

[Figure]

**Fig. R4. The time series of the standardized intensity of the upward motion over the tropical western Pacific (20°S-10°N, 100°E-180°E) at (a) 150 hPa; (b) 500 hPa; and (c) 700 hPa extracted from JRA55 (red), ERA5 (black) and MERRA2 (blue) datasets after removing the ENSO signal using linear regression method. The straight lines in each figure indicate the linear trends. The linear trends of the upward motion intensity over the TWP at 150 hPa, 500 hPa, and 700 hPa from three datasets are statistically significant at the 95% confidence level.**

P10 L201: 'This suggests a comprehensive enhancement of vertical velocity though the whole troposphere, which is evident from the surface to 100 hPa (not shown).' Figures demonstrating this could be shown in an electronic supplement.

**Re: The trends of vertical velocity from the surface to 100 hPa in NDJFM derived from JRA55, ERA5, and MERRA2 are added in the supplementary material (Supplementary Fig. 4)**

P10 L205 :'Due to the data limitation, it is not possible to show the corresponding changes of trace gases by observations.' I agree that it is difficult to find observation from 1958 to 2017. However satellite measurements from shorter time period could be used (e.g. MLS CO available since August 2004; https://mls.jpl.nasa.gov).

**Re: We thank for the referee's comment. An extra figure showing the trends of CO observed by MOPITT and MLS at around 200 hPa during 2000-2017 and 2005-2017 is added in the revised manuscript. The details could be found in the responses to the major comments above.**

P11 L210:   'of observed OLR' --> 'of observed OLR provided by NOAA (see Sect. 2)'

**Re: Corrected.**

P11 L222: 'CPTT' is not yet introduced. Fig. 4b is not referred to in the text  --> '.. the cold-point tropopause temperature (CPTT; see Fig. 4b) shows significantly decreasing trends over the TWP in NDJFM during 1958-2017,... However negative trends are also found in other regions in low and mid-altitudes, except in the Pacific.'

**Re: CPTT is introduced in the revised manuscript *Line 55*. The statement is**

**corrected.**

P12 L242: 'The SSTs over the TWP are positively correlated with the upward motion intensity over the TWP, while the SSTs over tropical central, eastern Pacific, and

Indian Ocean show negative correlations.' I am wondering that the positive correlation pattern is somewhat shifted to the east, then the western part of the maritime continent (100°E-120°E) is also negative correlated. However,  in the western part of the maritime continent (100°E-120°E) the trends of horizontal winds (Fig. 2) are large.

Maybe, it is useful to avoid misunderstandings to mark the region of the TWP

somehow (e.g. by a box).

**Re: We are sorry for the possible confusion. The TWP is marked by a box in the**

**figures of the revised manuscript, and the corresponding statement is corrected**

**to avoid the confusion.**

P13 L253:  'a couple of model simulations' -->    'a couple of model simulations with

WACCAM4'

**Re: Corrected.**

P14 L277: 'a couple of time-slice runs (R1 and R2) are performed (more details are given in the section 2).'   -->   It is maybe a matter of taste, but I would prefer in general to say 'simulations instead of 'run'. Please repeat the main features of R1 and

R2 as a reminder for the reader.

**Re: Corrected. And the main features of R1 and R2 are added to the corresponding paragraph.**

P14 L289: 'The changes in the OLR' --> 'The changes in the OLR simulated in WACCAM4'

**Re: Corrected.**

P15 L300: 'We now discuss about the relationship between the trends of the upward motion over the TWP and the changes of the trace gases in the lower stratosphere.' -->'The relationship between the trends of the upward motion over the TWP and the change of CO and water vapor in the lower stratosphere simulated with WACCAM4 will be analyzed. It is expected, that a positive trend in the upward motion over the TWP yield higher CO in the lower stratosphere caused be enhanced vertical upward transport. However, water vapor mixing ratios in the lower stratosphere depends in addition from the temperature in the UTLS ....' Is that what you would like to discuss here?

**Re: Yes. The corresponding phrases are corrected.**

Section 3.3 is written somewhat confusing, therefore I propose to write a short introduction of Sect. 3.3 summarizing previous results from the literature and subsequent the new results of Qie et al.

**Re: Thanks for the comment. A short introduction of Section 3.3 is added to the manuscript according to the comments of the referee and the literature.**

**"Previous studies showed that the enhanced deep convection and upward motion could lead to increased CO in the UTLS (e.g., Duncan et al., 2007; Livesey et al., 2013). At the same time, water vapor mixing ratios in the UTLS may increase due to the enhanced upward motion which could bring more wet air from low altitude to high altitude (e.g., Rosenlof, 2003; Lu et al., 2020). However, the water**

**vapor mixing ratios in the lower stratosphere also depend on the tropopause temperature (e.g., Highwood and Hoskins, 1998; Garfinkel et al., 2018; Pan et al., 2019). Hence, the relationship between the intensity of upward motion and the water vapor concentration in the UTLS is complex. Here, the relationship between the trends of the upward motion over the TWP and the changes in CO and water vapor in the ULTS simulated with WACCM4 are analyzed."**

P15 L303: 'in different simulations are displayed' --> 'are shown based on the Control and the Fixsst simulation as well as using their difference..'

**Re: Corrected.**

P15 L303: --> 'in Fig. 7d-i'

**Re: Corrected.**

P16 L328:   'As mentioned above in section 3.1, the observed tracer gases (e.g., CO) have very limited data record and may be affected by a mixture of anthropogenic and natural (e.g., biomass burning) emissions and the ENSO events (e.g., Duncan et al., 2007; Logan et al., 2008). It is therefore very hard to identify the relative contribution of single factors.' This sentence is here not very helpful, please remove it.

**Re: Removed.**

P16 L332: 'We utilize the numeric simulations' --> 'We use the Control and the Fixsst simulation with WACCAM4 ..'

**Re: Corrected.**

P17 L344:   'increasing trends over the TWP' How much is the increase in CO within 60 years? Please add some numbers in the text. ($4*10^{-4}$ ppm per year ->   0.024 ppm change in CO in 60 years; that seems not to be much.)

Give some reference about CO values and variability of CO in this region from
measurements to assess the trend in CO over TWP.

**Re: Thanks for the suggestion. We show the climatological mean CO values at**
**215 hPa in NDJFM from MLS observations during 2005-2017 and at 200 hPa in**
**NDJFM from MOPITT observations during 2000-2017. The concentration of**
**MLS CO over the TWP is approximately 80 ppbv at 215 hPa and MOPITT CO**
**is 70 ppbv at 200 hPa, which is consistent with previous study (e.g., Huang et al.,**
**2016). The increasing trends of CO at 150 hPa over the TWP in the Control and**
**Fixsst simulations are approximately 3.4 ppbv decade$^{-1}$ (20.4 ppbv within 60**
**years) and 3.2 ppbv decade$^{-1}$ (19.2 ppbv within 60 years). The CO at 150 hPa**
**over the TWP derived from the difference between the Control and Fixsst**
**increased 0.2 ppbv decade$^{-1}$ (1.2 ppbv within 60 years), which suggests that the**
**enhanced deep convection and intensified upward motion could lead to an extra**
**6% increasing trend of CO at 150 hPa over the TWP. It should be mentioned**
**that the changes in the CO at 150 hPa caused by the intensified upward motion**
**over the TWP not only depend on the vertical transport but also on the gradient**
**of CO concentration at around 150 hPa (Garfinkel et al., 2013). This may be the**
**reason why the intensifying upward motion over the TWP only contribute to an**
**extra 6% increasing trend of CO at 150 hPa in NDJFM during 1958-2017. For**
**example, CO derived from the difference between the Control and Fixsst**
**simulations shows higher increasing trends in the layer 150-70 hPa (0.4 ppbv**
**decade$^{-1}$) than those at 150 hPa (0.2 ppbv decade$^{-1}$), which is due to the greater**
**CO gradient in the UTLS comparing to the CO gradient in the upper**
**troposphere.**

P17 L354: 'This is consistent with our results which show intensified northerlies over
the subtropical Indian Ocean and strengthened westerlies over the subtropical Indian
Ocean and western Pacific'

Please add some numbers in the text: how much is the strengthening. Is it a large or
weak change. Please give the reader some numbers to assess this change.

**Re: Thanks for the suggestion. The trends of the northerlies over the subtropical**
**Indian Ocean (15°S-25°S, 60°E-100°E) are approximately 0.2 m s$^{-1}$ decade$^{-1}$ and**
**the trends of westerlies over the subtropical Indian Ocean and western Pacific**
**(20°N-35°N, 60°E-160°E) are approximately 0.3 m s$^{-1}$ decade$^{-1}$ (Figs. 5c and f).**
**The discussion is added to the revised manuscript.**

P18 L377: 'In summary, the increase of CO as shown in Figs. 8a-8b is mainly caused
by surface emissions.' My understanding is that the surface emissions are the same in
the Control and Fixsst simulation and that the increase of UTLS CO is caused by
stronger upwelling. Please clarify.

**Re: We are sorry for the confusion. The surface emissions are the same in the**
**Control and Fixsst simulations, which are increasing in NDJFM during**
**1958-2017. Hence, the trends of CO in Fig. 9a (in the revised manuscript) contain**
**the CO trends induced both by the increased surface emissions and the enhanced**
**upward motion. The trends of CO over the TWP in Fig. 9b (in the revised**
**manuscript) only include the CO trends induced by the increased surface**
**emissions since the upward motion over the TWP in the Fixsst simulation shows**
**weak trends. Furthermore, the CO increased through the troposphere over the**
**TWP using the difference between the Control and Fixsst simulations, which**
**suggests that the increase of CO in the upper troposphere in Fig. 9c (in the**
**revised manuscript) is caused by the intensified upward motion over the TWP.**
**Some discussions are added to the text.**

Figure 11: '(a) Control run; (b) Fixsst run; (c) difference between the Control run and
the Fixsst run; and (d) JRA55.'   --> labels a,b,c,d are not consistent to Fig.11.

**Re: We are sorry for the mistake. The figure caption is corrected.**

Why is MERRA2 and ERA5 not shown. How is the trend of the BD circulation calculated? Are zonal mean values shown? Please clarify.

**Re: Thanks for the suggestion. We have added the trends of the BDC derived**

**from ERA5 and MERRA2 to the supplementary material. The trend of the BDC**

**is calculated using the simple least square regression. The $w^*$ used in the**

**manuscript is calculated using the TEM formula and $w^*$ denotes the monthly**

**zonal mean of the vertical component of the BDC. To avoid confusion, the $\overline{w}^*$**

**and $\overline{v}^*$ in the equation mentioned in the original manuscript are corrected as**

**$w^*$ and $v^*$ in the revised manuscript.**

P19 L384: 'The tropical upwelling of BDC (w*) are significantly increased in the lower stratosphere over past decades as seen in both reanalysis data and the control run (Figs. 11a and b).'   --> 'in JARA55 and control simulation'

**Re: Corrected.**

Please indicate that the TEM is used to calculate w*. Please specify 'significantly increased' with some numbers. Please compare the increase with numbers from other references.

**Re: We thank the referee's comment. The manuscript is revised correspondingly.**

**The quantitative results and the comparison with other references are added.**

**The tropical upwelling of BDC ($w^*$) calculated using the TEM formula increased**

**significantly in the lower stratosphere over past decades as seen in the JRA55**

**data and the Control simulation (Figs. 12a and 12b). We found that the 70 hPa**

**upward mass flux in NDJFM in the tropics (15°S-15°N) increased 2.8±1.9%**

**decade$^{-1}$ ( significant at the 95% confidence level) in the JRA55 data from 1958**

to 2017 (Fig. 12a) and 4.6±4.3% decade$^{-1}$ ( significant at the 95% confidence level)

in the MERRA2 data from 1980 to 2017 (Supplementary Fig. 7b). From the

ERA5 data, the 70 hPa upward mass flux in NDJFM increased in the north hemisphere (0-15°N) at a rate of 5±2.8% decade$^{-1}$ ( significant at the 95%

confidence level), but decreased significantly in the south hemisphere (0-15°S)

during 1958-2017 (Supplementary Fig. 7a). On average, the trend of the 70 hPa upward mass flux in NDJFM in the tropics (15°S-15°N) is insignificant in ERA5.

In fact, many previous studies have investigated the trends of BDC. For example,

Abalos et al. (2015) investigated the trends of BDC using JRA55, MERRA, and

ERA-Interim data during 1979-2012 and suggested that the BDC in JRA55 and

MERRA significantly strengthened throughout the layer 100-10 hPa with a rate of 2-5% decade$^{-1}$, while the BDC in ERA-Interim shows weakening trends. Diallo et al. (2021) compared the trends of the BDC in the ERA5 and ERA-Interim during 1979-2018 and pointed out that the BDC in the ERA-Interim shows weakening trend and the BDC in the ERA5 strengthened at a rate of 1.5%

decade$^{-1}$ which is more consistent with other studies. In the present study, we only focus on the trend of the BDC in the wintertime (NDJFM) in the tropics (15°S-15°N) during 1958-2017, which may lead to some differences between our result and the previous studies. Overall, the trends of the tropical upwelling of

BDC using JRA55, MERRA2 data and the Control simulation are similar to the previous studies using both reanalysis datasets and model results (e.g., Butchart et al., 2010; Abalos et al., 2015; Fu et al., 2019; Rao et al., 2019; Diallo et al.,

2021). However, the tropical upwelling of the BDC decreased using ERA5 data in the tropics (15°S-15°N), which are different from the results in JRA55 and

MERRA2. In summary, the tropical upwelling of the BD circulation is likely strengthened as shown in JRA55 and MERRA2 reanalyses as well as model simulations, although there are some uncertainties since the ERA5 data show a negative trend. This may contribute to the transport of the tropospheric trace gases from the TTL to a higher level. The increased concentration of CO in the

UTLS in Fig. 9c and 10f may be due to a combined effect of the strengthened

**tropical upwelling of the BD circulation and the enhanced upward motion over**

**the TWP.**

[Figure]

**Fig. R5. The trends of the BD circulation (vectors) calculated using the TEM**

**formula using ERA5 and MERRA2 data. (a) The trends of $w^*$ ($10^{-5}$ m s$^{-1}$ a$^{-1}$) and**

**$v^*$ ($10^{-2}$ m s$^{-1}$ a$^{-1}$) in NDJFM during 1958-2017 using ERA5 data. (b) The trends**

**of $w^*$ ($10^{-5}$ m s$^{-1}$ a$^{-1}$) and $v^*$ ($10^{-2}$ m s$^{-1}$ a$^{-1}$) in NDJFM during 1980-2017 using**

**MERRA2 data. The shadings are the trends of the vertical velocities ($10^{-5}$ m s$^{-1}$**

**a$^{-1}$). The trends of the vertical velocity over the dotted regions are statistically**

**significant at the 90% confidence level.**

P19 L400: 'The recent trends of the upward motion from the lower to the upper troposphere in boreal winter over the TWP is investigated for the first time based on the reanalysis datasets and model simulations.' Specify which reanalysis and which model runs are used.

**Re: Corrected.**

P19 L405: 'Warmer SSTs over the TWP lead to a strengthened Pacific Walker
circulation, enhanced deep convection and stronger upward motion over the TWP.'
Please make this statement more quantitative. From the analysis it is not clear for me
what is enhanced: convection or subsequent upward motion over the TWP by diabatic
heating or both.

**Re: Thanks for the suggestion. The statement is rephrased. Both of the deep**
**convection and the subsequent upward motion over the TWP by diabatic heating**
**are enhanced. We are sorry for the confusion.**

How is downward transport over TWP by the Pacific Walker circulation during El
Niño considered within the analysis? Please clarify?

**Re: Thanks for the comment. The impact of ENSO events on the upward motion**
**over the TWP is discussed in the revised manuscript according to the referee's**
**suggestion. Some discussions are also added in the Summary and Discussion.**

P20 L410:' Model simulations indicate that the CO concentration increases
significantly from the surface to the stratosphere with increased surface emissions.'
Please make the statement more quantitative.

**Re: Thanks for the comment. The statement is rephrased as: "Results from the**
**Control simulation indicate that the CO concentration increased significantly**
**from the surface to the stratosphere over the TWP. The CO at 150 hPa increased**
**at a rate of approximately 3.4 ppbv decade$^{-1}$ with increased surface emissions**
**and the enhanced upward motion over the TWP. Specifically, an enhancement of**
**tropospheric upward motion and subsequent upward transport of trace gases**
**over the TWP lead to an extra 6% increasing trend of CO concentrations in the**
**upper troposphere. Furthermore, the upward mass fluxes at 70 hPa in the**
**tropics (15°S-15°N) show strengthening trends at rates of 2.8±1.9% decade$^{-1}$ and**
**4.6±4.3% decade$^{-1}$ in JRA55 data (during 1958-2017) and MERRA2 data (during**

**1980-2017), respectively, which is consistent with previous studies (e.g., Butchart**

**et al., 2010; Fu et al., 2019; Rao et al., 2019).”**

P20 L417: 'Trace gases and aerosols in the stratosphere have important impacts on the stratospheric processes, and hence influence the troposphere weather and climate through their radiative and dynamical feedback'. This statement is very general.

Please be more specific here.

**Re: We thank the referee's comment. The statement is rephrased as: "Trace**

**gases and aerosols entering the stratosphere from the troposphere have**

**important impacts on the stratospheric processes. For example, ozone-depleting**

**substances, $CH_4$ and $N_2O$ could influence on the stratospheric ozone significantly**

**(e.g., Shindell et al., 2013; Wang et al., 2014; WMO, 2018), which also modify the**

**temperature in the stratosphere significantly through their strong radiative**

**effects. Water vapor in the lower stratosphere, in particular, has a significant**

**warming effect on the surface climate (Solomon et al., 2010). Therefore, changes**

**of trace gases in the UTLS have important impacts on both tropospheric and**

**stratospheric climate.”**

My impression is that the conclusion section should be revised to summarize the results of Qie et al in a much more quantitative way.

**Re: Thanks for the referee's suggestion. The conclusion section is revised**

**according to the quantitative results in the revised manuscript.**

**The conclusion section is rewritten as:**

[revised manuscript text omitted]

---

## Author Comment (AC2)

**Responses to the comments by Referee #2**

**Manuscript number**: acp-2021-647

**Title**: **Enhanced upward motion through the troposphere over the tropical**
**western Pacific and its implications for the transport of trace gases from the**
**troposphere to the stratosphere**

**Author(s)**: Kai Qie, Wuke Wang, Wenshou Tian[*], Rui Huang, Mian Xu, Tao Wang,

Yifeng Peng

**December 2021**

The manuscript presents an analysis of atmospheric upward transport through the upper troposphere and lower stratosphere over the tropical West Pacific based on reanalysis data and model observations. Long-term changes in the upwelling are linked to increasing global sea surface temperatures leading to a strengthening of the Pacific Walker circulation and deep convection. Implications for stratospheric entertainment of CO and H2O are discussed.

The research question addressed here is an important one and the topic is of general interest to the readers of ACP. Some parts of the analysis are solid and provide valuable insights into long-term changes of the underlying processes. However, I have some major concerns (listed below) and recommend major revisions before the manuscript can be published.

**Re: We thank for the reviewer's helpful comments. We have revised the manuscript thoroughly according to the comments and the manuscript has been improved substantially. The point-to-point responses are listed below.**

**Major comments**

1) Caution is advised when using reanalysis data for trend detection as the quality and character of reanalyses may have changed over time and non-physical trends can result from changes in the observing system or execution stream. This has been demonstrated for many atmospheric quantities such as stratospheric temperature (Long et al., 2017, ACP) and residual circulation velocities (Chapter 5, S-RIP report, 2021).

Here, the trends derived from reanalysis are presented without any discussion of these aspects, but instead are used as if they would be reliable sources of long-term changes. A discussion of the limitations of reanalysis data for trend studies and words of caution are needed and the text should be changed accordingly throughout the manuscript, in particular when using reanalysis before 1979.

Re: We thank the reviewer for the very important comment. We totally agree with the reviewer that the limitations of reanalysis data for trend analysis should be discussed. Such discussion is added to the Section 2.

The text has been revised as: "A special caution is needed because of the limitations of reanalysis data. The reanalysis datasets assimilate observational data based on the ground- and space-based remote sensing platforms to provide more realistic data products. However, previous studies suggested that there are still uncertainties in the reanalysis data (e.g., Simmons et al., 2014; Long et al.,

2017; Uma et al., 2021). The accuracy of the vertical velocity in reanalysis data sets has been evaluated by the Reanalysis Intercomparison Project (Fujiwara et al., 2017), which is initiated by the Stratosphere-troposphere Processes And their

Role in Climate (SPARC). Results of a comparison between the radar observed data and the reanalysis data indicate that the updrafts in the UTLS are captured well near the TWP even though there are still large biases in the reanalysis datasets and the updrafts from the JRA55 data are stronger than those from the

ERA5 and MERRA2 data (Uma et al. 2021). Additionally, discontinuities in the reanalysis data due to different observing systems (for example, transition from

TOVS to ATOVS) may still exist (e.g., Long et al., 2017), which could lead to uncertainties in the long-term trend of a certain meteorological filed. Hitchcock (2019) suggested that the reanalysis uncertainty is larger in the radiosonde era (after 1958) than in the satellite era (after 1979), but the radiosonde era is of equivalent value to the satellite era because the dynamical uncertainty dominates in the both eras. The data in the radiosonde era (1958-1978) used in the present study may induce uncertainties in our results. Therefore, we discuss the trends for both the periods of 1958-2017 and 1980-2017. In addition, we combine three most recent reanalysis datasets (JRA55, ERA5, and MERRA2) to obtain relatively robust results."

The description about the trend analysis is also revised accordingly throughout the manuscript.

2) Trends of the vertical wind derived from the three reanalysis data sets agree in some regions but disagree in others as seen from Figure 2. A discussion of the level of agreement is needed. At the same time, it is not clear which region exactly is referred to as the tropical western Pacific (TWP). In many cases the authors would us the TWP

in cases when the text and figures suggest that they refer to the Maritime Continent (e.g., ERA5 shows increasing trend of w over the Maritime Continent but decreasing trends over larger parts of the TWP). It would be very helpful, if the authors would define the regions upfront and use them consistently throughout the manuscript.

**Re: Thanks for the comment. Some discussions about the trends of horizontal**

**winds and vertical velocity in the JRA55, ERA5, and MERRA2 are added to the**

**revised manuscript. The differences between the reanalysis datasets may be**

**mainly due to the different time periods which are used to calculate the linear**

**trends in JRA55 (1958-2017), ERA5 (1958-2017) and MERRA2 (1980-2017). An**

**additional figure showing the trends of horizontal winds and vertical velocity in**

**the JRA55, ERA5, and MERRA2 (Fig. R1) during 1980-2017 is added to the**

**supplementary material (Supplementary Fig. 3). The discussion in the revised**

**manuscript is expressed as:**

**"Such an enhancement of the upward motion over the TWP is evident in all**

**three reanalysis datasets used here (JRA55, ERA5, and MERRA2), although**

**there are also some differences between the three reanalysis datasets. For**

**example, the trends of the horizontal winds in the upper troposphere in**

**MERRA2 (Fig. 2c) are larger than those in JRA55 and ERA5 (Figs. 2a and b).**

**There are negative trends of vertical velocity in JRA55 and ERA5 while positive**

**trends of vertical velocity in MERRA2 over the northern Pacific (Figs. 2a-c).**

**However, these differences are mainly due to the different time periods used to**

**calculate the linear trends in JRA55 (1958-2017), ERA5 (1958-2017) and**

**MERRA2 (1980-2017). Supplementary Fig. 3 gives the trends of $w$ and**

**horizontal winds in NDJFM during 1980-2017 using JRA55, ERA5, and**

**MERRA2 data, which shows insignificant differences between these reanalysis**

**datasets. The trend patterns of the horizontal winds in JRA55, ERA5, and**

**MERRA2 are consistent with each other (Supplementary Fig. 3). For the trends**

**of vertical velocity, significant positive trends over the TWP region can be noted**

**in the JRA55, ERA5, and MERRA2 datasets, although the trends in ERA5 are**

**slightly weaker than those in JRA55 and MERRA2 (Fig. 2 and Supplementary**

**Fig. 3). Comparing to the negative trends of the vertical velocity over the central**

**Pacific in JRA55 and ERA5, the negative trends in MERRA2 extend more**

**northward (Supplementary Fig. 3)."**

**The TWP region is defined as 20°S-10°N, 100°E-180°E. According to the**

**referee's comment, the TWP is marked using a black rectangle in the figures of**

**revised manuscript.**

[Figure]

**Fig. R1. The trends of the vertical velocity and horizontal winds in NDJFM using**

**JRA55 (a, d, g), ERA5(b, e, h) and MERRA2(c, f, i) data during 1980-2017 at**

**different levels. (a)-(c) are the trends of winds at 150 hPa. (d)-(f) are the trends of**

**winds at 500 hPa. (g)-(i) are the trends of winds at 700 hPa.**

3) It seems that the upwelling trends (averaged over the region of interest) are hardly significant even at the 90% confidence level. The uncertainty ranges and trend values need to be provided in the text or figure. Furthermore, it is not clear why the averaging is done over 20S-10N. Looking at Figure 2, my impression is the averaging over 20S-20N will not result in trends significant at the 90% confidence level. If this is the case, it should be stated in the text.

**Re: The uncertainty ranges and trend values are shown in the revised**

**manuscript. "The intensity of the upward motion over the TWP at 150 hPa**

**increased $3.0\pm1.2\times10^8$ kg s$^{-1}$ decade$^{-1}$ ($8.0\pm3.1\%$ decade$^{-1}$), $1.3\pm1.2\times10^8$ kg s$^{-1}$**

**decade$^{-1}$ ($3.6\pm3.3\%$ decade$^{-1}$), and $3.0\pm2.8\times10^8$ kg s$^{-1}$ decade$^{-1}$ ($7.5\pm7.1\%$ decade$^{-1}$)**

**in JRA55, ERA5, and MERRA2 data, respectively. As shown in Figs. 3b and c,**

**the intensity of the upward motion at 500 hPa and 700 hPa in JRA55 and the**

**intensity of the upward motion at 500 hPa in ERA5 over the TWP also increased**

**significantly at 95% confidence level ($4.6\pm2.6\times10^8$ kg s$^{-1}$ decade$^{-1}$, $2.9\pm1.7\times10^8$ kg**

**s$^{-1}$ decade$^{-1}$, and $2.5\pm2.5\times10^8$ kg s$^{-1}$ decade$^{-1}$, respectively). The increasing trends**

**of the intensity of the upward motion at 700 hPa in ERA5 and at 500 hPa and**

**700 hPa in MERRA2 are significant at the 90% confidence level at rates of**

**$1.9\pm1.6\times10^8$ kg s$^{-1}$ decade$^{-1}$, $5.4\pm5.3\times10^8$ kg s$^{-1}$ decade$^{-1}$ and $3.9\pm3.8\times10^8$ kg s$^{-1}$**

**decade$^{-1}$, respectively. "**

**The description about how to calculate the uncertainty ranges is also added to**

**the Section 2 as:**

**"The linear trends are estimated using a simple least square regression method.**

**The significances of the correlation coefficients, mean differences, and trends are**

**determined via a two-tail Student's t-test. The confidence interval of trend is**

**calculated using the following equation (Shirley et al., 2004):**

$$\left( b - t_{1-\frac{\alpha}{2}}(n-2)\sigma_b, b + t_{1-\frac{\alpha}{2}}(n-2)\sigma \right)$$

**where b is the estimated slope, $\sigma$ denotes the standard error of the slope, and**

**$t_{1-\frac{\alpha}{2}}(n-2)$ represents the value of t-distribution with the degree of freedom**

equal to **n-2**. **α** is the two-tailed confidence level. $\sigma$ is calculated as:

$$\sigma = b\sqrt{\frac{\frac{1}{r^2}-1}{n-2}} \text{.''}$$

The averaging is done over 20°S-10°N because of two reasons: 1. The center of upward motion in the boreal winter (NDJFM) over the tropical western Pacific is mainly located in the region over 20°S-10°N. 2. The intensification of upward motion over the tropical western Pacific is more significant over 20°S-10°N. To avoid confusion, some explanations are added to the revised manuscript.

The confidence level of significance of the trend analysis could be impacted by the fluctuations in the time series. The other referee pointed out that there are extreme minima in the time series of the upward motion over the TWP (Fig. 3), which are mainly due to the ENSO events. Here, the time series of the upward motion over the TWP with the ENSO signal removed using the single linear regression method are also shown (Fig. R2). It could be seen that the extreme minima become much weaker after removing the ENSO signal using the linear regression method. This result suggests that the El Niño events could affect the upward motion over the TWP and to a large extent result in the extreme minima (1982, 1991, and 1997). After removing the large fluctuations due to the ENSO

events, the upward motions over the TWP at 150 hPa, 500 hPa, and 700 hPa in

NDJFM in JRA55, ERA5, and MERRA2 show statistically significant intensifying trends above the 95% confidence level.

[Figure]

**Fig. R2. The time series of the standardized intensity of the upward motion over the tropical western Pacific (20°S-10°N, 100°E-180°E) at (a) 150 hPa; (b) 500 hPa; and (c) 700 hPa extracted from JRA55 (red), ERA5 (black) and MERRA2 (blue) datasets after removing the ENSO signal using linear regression method. The straight lines in each figure indicate the linear trends. The linear trends of the upward motion intensity over the TWP at 150 hPa, 500 hPa, and 700 hPa from three datasets are statistically significant at the 95% confidence level.**

4) Where is the cold point temperature trend coming from (Figure 4)? This data source is not listed in the text or caption. Given that it starts at 1958, most likely the trend is derived from JRA55. Again, some words of caution are needed, given that cold point temperature trends from reanalysis data sets can show significant differences even for the satellite period (Tegtmeier et al., 2020, ACP).

**Re: We thank for the referee's comment. The trend of CPTT in Fig. 4 is from JRA55 data. The data source is added to the figure caption in the revised manuscript. Caution is added to the revised manuscript as: "It should be noted that the CPTT from different reanalysis datasets may show different trends even**

**for the satellite period (Tegtmeier et al., 2020). Additionally, the JRA55 data**

**before 1978 may also lead to uncertainties in the CPTT trends. Caution is needed**

**when discussing the trends of CPTT from reanalysis datasets."**

5) The discussion of the trends of stratospheric upwelling needs to refer to Chapter 5

of the SPARC S-RIP report. Chapter 5 states in its abstract: 'However, estimates of long-term trends in tropical upwelling are inconsistent among different products, showing either strengthening, weakening, or no trend.' Therefore, results shown in

Figure 11 based on JRA55 are most likely not consistent with other reanalyses.

**Re: We thank the referee's comment. The discussion of the trends of**

**stratospheric upwelling is rewritten. The trends of stratospheric upwelling in**

**ERA5 and MERRA2 are added to the supplementary material (Fig. R3). The**

**discussion is written as:**

**"The tropical upwelling of BDC ($w^*$) which calculated using the TEM**

**formula increased significantly in the lower stratosphere over past decades as**

**seen in the JRA55 data and the Control simulation (Figs. 12a and 12b). We found**

**that the 70 hPa upward mass flux in NDJFM in the tropics (15°S-15°N)**

**increased 2.8±1.9% decade$^{-1}$ ( significant at the 95% confidence level) in the**

**JRA55 data from 1958 to 2017 (Fig. 12a) and 4.6±4.3% decade$^{-1}$ ( significant at**

**the 95% confidence level) in the MERRA2 data from 1980 to 2017**

**(Supplementary Fig. 7b). From the ERA5 data, the 70 hPa upward mass flux in**

**NDJFM increased in the north hemisphere (0-15°N) at a rate of 5.0±2.8%**

**decade$^{-1}$ (significant at the 95% confidence level), but decreased significantly in**

**the south hemisphere (0-15°S) during 1958-2017 (Supplementary Fig. 7a). On**

**average, the trend of the 70 hPa upward mass flux in NDJFM in the tropics**

**(15°S-15°N) is insignificant in ERA5. In fact, many previous studies have**

**investigated the trends of BDC. For example, Abalos et al. (2015) investigated the**

**trends of BDC using JRA55, MERRA, and ERA-Interim data during 1979-2012**

**and suggested that the BDC in JRA55 and MERRA significantly strengthened**

**throughout the layer 100-10 hPa of order 2-5% decade$^{-1}$, while the BDC in**

ERA-Interim shows weakening trends. Diallo et al. (2021) compared the trends of the BDC in the ERA5 and ERA-Interim during 1979-2018 and pointed out that the BDC in the ERA-Interim shows weakening trend and the BDC in the

ERA5 strengthened 1.5% decade$^{-1}$ which is more consistent with other studies. In the present study, we only focus on the trend of the BDC in the wintertime (NDJFM) in the tropics (15°S-15°N) during 1958-2017, which may lead to some differences between our result and that in the previous studies. Overall, the trends of the tropical upwelling of BDC derived from JRA55, MERRA2 data and the Control simulation are similar to that in previous studies using both reanalysis datasets and model results (e.g., Butchart et al., 2010; Abalos et al.,

2015; Fu et al., 2019; Rao et al., 2019; Diallo et al., 2021). However, the tropical upwelling of the BDC decreased in ERA5 data in the tropics (15°S-15°N), which are different from the results in JRA55 and MERRA2. ”

“In summary, the tropical upwelling of the BDC is likely strengthened as shown in JRA55 and MERRA2 reanalyses as well as model simulations, although there are some uncertainties since the ERA5 data show a negative trend. This may impact on the transport of the tropospheric trace gases from the TTL to a higher altitude. The increased concentration of CO in the UTLS in Fig. 8c and 10f may be due to a combined effect of the strengthened tropical upwelling of the BD

circulation and the enhanced upward motion over the TWP.”

[Figure]

**Fig. R3. The trends of the BD circulation calculated using the TEM formula in ERA5 and MERRA2. (a) The trends of w\* ($10^{-5}$ m s$^{-1}$ a$^{-1}$) and v\* ($10^{-2}$ m s$^{-1}$ a$^{-1}$) in NDJFM during 1958-2017 using ERA5 data. (b) The trends of w\* ($10^{-5}$ m s$^{-1}$ a$^{-1}$) and v\* ($10^{-2}$ m s$^{-1}$ a$^{-1}$) in NDJFM during 1980-2017 using MERRA2 data.**

6) I don't agree with the interpretation the CO changes based on various model runs
as presented in Figure 9. Both simulations have the same sources and the control run
shows enhanced convective uplifting brining more CO to higher altitudes. For the
tropical West Pacific, the trends are larger for the Control run throughout the whole
vertical extent of the troposphere. However, enhanced upwelling would result in a less
strong trend at the surface and boundary layer, opposite to what the simulations
indicate here. In fact, some recent studies showed that over the Indian Ocean, CO
abundance in the boundary layer decreases (despite the growing sources) while it
increases in the mid to upper troposphere due to enhanced convective activity (e.g.,
Girach and Nair, 2014). The discussions and conclusions regarding this figure need to
be revised.

**Re: We thank for the referee's comment. According to the referee's comment,**
**the reason for the increasing trends of CO in the lower troposphere shown in Fig.**
**9f is further investigated. The trends of CO in the lower troposphere using the**

Control and Fixsst simulations as well as the difference between them are shown (Fig. R4). The trends of difference of horizontal winds at 925 hPa between the

Control and Fixsst simulations are also shown (Fig. R4c). It can be found that there are northerly trends over east Asia and northeasterly trends near the south

Asia (Fig. R4c), which suggests that more CO-rich air from east Asia and south

Asia could be transported to the TWP in the Control simulation comparing to the Fixsst simulation. Since the CO concentration at 900 hPa over the northern

Pacific is higher than that over southern Pacific (Fig. R5), the northerly trends over the western and central Pacific may also contribute to the increased CO in the lower troposphere over the TWP in Fig. 9f. The interpretation about the Fig.

9 is revised in the revised manuscript as:

"It should be mentioned that the increasing trends of CO in the lower troposphere in Fig. 10f may be mainly caused by the changes in the horizontal winds. Girach and Nair (2014) suggested that enhanced deep convection and the subsequent intensified upward motion may lead to a decreased CO

concentration in the lower troposphere and an increased CO concentration in the upper troposphere. The trends of horizontal winds at 925 hPa are shown in

Supplementary Fig. 8c. There are northerly trends over east Asia and northeasterly trends near the south Asia (Supplementary Fig. 8c), which suggests that more CO-rich air from east Asia and south Asia could be transported to the

TWP in the Control simulation comparing to the Fixsst simulation. Since the CO

concentration in the lower troposphere over the northern Pacific is higher than that over southern Pacific, the northerly trends over the western and central

Pacific may also contribute to the increased CO in the lower troposphere over the TWP in Fig. 10f."

[Figure]

**Fig. R4. The trends of CO ($10^{-4}$ ppmv) at 925 hPa in NDJFM during 198-2017 in the (a) Control simulation, (b) Fixsst simulation, and (c) the difference between the Control and Fixsst simulations. The vectors in (c) denote the trends of the difference of 925 hPa horizontal winds ($10^{-1}$ m s$^{-1}$) between the Control and Fixsst simulations.**

[Figure]

**Fig. R5. The climatological mean CO concentration at 900 hPa in NDJFM during 2000-2017 using MOPITT data.**

**Minor comments**

Should the title say '… implications for …'?

**Re: Corrected.**

For the fact that halogenated gases are enhanced over the WP, a citation is needed.
The citations given at the end refer to tropospheric halogen chemistry. What is meant
with the second part of the sentence? A general statement, that halogens impact
stratospheric ozone chemistry? Or that halogens injected over the West Pacific have a
relatively large impact on stratospheric ozone chemistry?

**Re: We thank for the referee's comment. Citations are added to the revised**
**manuscript. The sentence is rewritten according to this comment and the**
**comment of the other referee as:**

**"Through the TWP region, tropospheric trace gases, e.g., the natural maritime**
**bromine-containing substances and outflow from anthropogenic emissions from**
**South Asia, are lifted to the upper troposphere and lower stratosphere (UTLS)**
**by the strong upward motion and the deep convection and subsequently into the**
**stratosphere by the large-scale upwelling (e.g., Levine et al., 2007, 2008; Navarro**
**et al., 2015), which affects the ozone concentration and other chemical processes**
**in the stratosphere (e.g., Feng et al., 2007; Sinnhuber et al., 2009)."**

Line 190: What is an intensifying trend? A trend increasing over time?

**Re: Sorry for the confusing. It should be a positive trend, not an intensifying**
**trend. We have corrected the sentence in the revised manuscript.**

Line 272: figure 2f shows wind fields at 500 hPa. Do you mean a different figure
here?

**Re: We are sorry for the mistake. It should be Figure 4d here. The mistake is**
**corrected in the revised manuscript.**

Line 270-274: This line of argumentation doesn't make any sense to me, and it is not
clear what the authors are trying to say.

**Re: We are sorry for the confusion. The sentence is rewritten as:**

[revised manuscript text omitted]

---

## Referee Report (RR1)

Second review of

Enhanced upward motion through the troposphere over the tropical western Pacific and its implications for the transport of trace gases from the troposphere to the stratosphere

by Qie et al.

The authors have adequately responded to my concerns given in the first review. In the revised version of the manuscript, the authors have 1) added a discussion of the limitations of the analysis, 2) added uncertainty ranges to trend values and extended the corresponding discussion, 3) corrected the physical mechanisms for the CO trends and 4) addressed all my minor comments.

---

## Author Response (AR2)

**Responses to the comments by Referee #2**

**Manuscript number**: acp-2021-647

**Title**: **Enhanced upward motion through the troposphere over the tropical**

**western Pacific and its implications for the transport of trace gases from the**

**troposphere to the stratosphere**

**Author(s)**: Kai Qie, Wuke Wang[*], Wenshou Tian, Rui Huang, Mian Xu, Tao Wang,

Yifeng Peng

**March 2022**

The revised version of the manuscript is much improved compared to the original manuscript and the authors did a lot of efforts to follow the reviewers advice. I recommend to publish the manuscript after some minor revisions.

**Re: We thank for the reviewer's detailed and helpful comments. We have revised the manuscript according to the comments. The point-to-point responses are listed below.**

1) Figures:

Most figures consist of different panels displaying the same x-y-ranges (e.g. Fig. 1, 2, 5, 6, 7, 9, 10, 11 and 12). To enhance the visibility and size of each panel I strongly recommend to remove x- and y-labels of some of the panels (e.g. display only x-label in bottom row and y-labels in the left column), panel titles as well as the white space between the panels.

**Re: Thanks for the comments. We have updated all the figures as suggested in the revised manuscript.**

3) Figures captions:

In the figure caption of several figures the text is very similar (Fig. 1, 5, 7 or Fig. 10, 11). There is a lot of repetition and the captions are a bit boring to read. I recommend to shorten some of the captions and use formulations such as 'Fig.2: The same as Fig.1, but...' to emphasize more the difference between the different figures. That would help to direct the reader's attention more to the main message of the figure.

**Re: Thanks for the suggestions. We have updated the figure captions as suggested.**

4) P7 Reanalysis data:

Maybe the authors could add a table summarizing the characteristics of the three reanalyses JAR55, ERA5, MERRA2 displaying horizontal, vertical and time resolution, considered time period etc. The text could then be shortened accordingly.

**Re: Thanks for the comments. The text in the revised manuscript is shortened.**

**And a table is added as:**

**Table 1. Basic specifications of JRA55, ERA5, and MERRA2 used in this study.**

| Name | Organization | Time period | Spatial resolution | Temporal resolution | Data assimilation |
|---|---|---|---|---|---|
| JRA55 | JMA | 1958-present | 55 km; L60 | 6-hourly | 4D-Var |
| ERA5 | ECMWF | 1950-present | 31 km; L137 | hourly | 4D-Var |
| MERRA2 | NASA GMAO | 1980-present | 0.5°×0.625°; L72 | 3-hourly | 3D-Var |

5) P9 L165: Please add a sentence (and some citations) why CO is a useful tropospheric tracer e.g. mention its lifetime.

**Re: Thanks for the helpful suggestion. A sentence is added to the revised**

**manuscript as:**

**"Since CO has a photochemical lifetime in the range of 2-3 months (Xiao et al.,**

**2007), it could be utilized as a tracer of cross-region transport in the troposphere**

**and the lower stratosphere (Park et al., 2009). Here, CO is used as a tropospheric**

**tracer to indicate the vertical transport from the near-surface to the upper**

**troposphere and the lower stratosphere."**

6) P10 L190:

Also here a table summarizing the different features of the performed model runs would be very helpful to obtain an better overview and as a quick reminder when reading the results and summary section.

**Re: Thanks for the comments. A table is added to the revised manuscript as:**

**Table 2. Description of simulations with WACCM4.**

| Experiment | Description |
|---|---|
| Control | Transient simulation. Observed greenhouse gases and solar irradiances. Prescribed SST forcing using observed SST. |
| Fixsst | Transient simulation. Observed greenhouse gases and solar irradiances. Prescribed SST forcing using monthly mean climatology from 1958 to 2017. |
| R1 | Time-slice simulation. SSTs prescribed as the climatological mean of 1998-2017 over the region 20°S-20°N, 120°E-160°E in NDJFM, but fixed as climatological mean of 1958-2017 over other regions. |
| R2 | Same as R1, but the SSTs over the region 20°S-20°N, 120°E-160°E are prescribed as the climatological mean SSTs during 1958-1977. |

7) P20 Summary and discussion:

Also here for better comparability, I strongly recommend to summarize the results in a table showing the increase of upward motion resulting from the the three different reanalyses, from WACCM4 as well as the CO increase.

**Re: Thanks for the comments. A table is added to the revised manuscript as:**

**Table 3. The trends of the upward motion over the TWP at 150 hPa, 500 hPa, and 700 hPa in NDJFM during 1958-2017 from JRA55, ERA5, MERRA2, Control simulation and Fixsst simulation. And the trends of 150 hPa CO from the Control and Fixsst simulations.**

| | JRA55 | ERA5 | MERRA2 | Control | Fixsst |
|---|---|---|---|---|---|
| **150 hPa Upward motion** | $3.0\pm1.2\times10^9$ kg s$^{-1}$ decade$^{-1}$ | $1.3\pm1.2\times10^9$ kg s$^{-1}$ decade$^{-1}$ | $3.0\pm2.8\times10^9$ kg s$^{-1}$ decade$^{-1}$ | $2.0\pm1.2\times10^9$ kg s$^{-1}$ decade$^{-1}$ | $-4.8\pm6.4\times10^8$ kg s$^{-1}$ decade$^{-1}$ |
| **500 hPa** | $4.6\pm2.6\times10^9$ | $2.5\pm2.5\times10^9$ | $5.4\pm5.3\times10^9$ | $3.5\pm2.4\times10^9$ | $-1.0\pm1.3\times10^9$ |

| Upward motion | kg s$^{-1}$ decade$^{-1}$ | kg s$^{-1}$ decade$^{-1}$ | kg s$^{-1}$ decade$^{-1}$ | kg s$^{-1}$ decade$^{-1}$ | kg s$^{-1}$ decade$^{-1}$ |
|---|---|---|---|---|---|
| 700 hPa Upward motion | $2.9\pm1.7\times10^{9}$ kg s$^{-1}$ decade$^{-1}$ | $1.9\pm1.6\times10^{9}$ kg s$^{-1}$ decade$^{-1}$ | $3.9\pm3.8\times10^{9}$ kg s$^{-1}$ decade$^{-1}$ | $1.8\pm1.4\times10^{9}$ kg s$^{-1}$ decade$^{-1}$ | $-6.3\pm8.1\times10^{8}$ kg s$^{-1}$ decade$^{-1}$ |
| 150 hPa CO | -- | -- | -- | 3.4 ppbv decade$^{-1}$ | 3.2 ppbv decade$^{-1}$ |

8) P31 L651:

'More observational data are expected to be used to obtain a more robust result in the future.' This sentence sounds odd, please revise it. --> 'The availability of more high resolution observations in the future could maybe enhance the quality of the reanalyses data.' Is that what the authors would like to point out here?

**Re: The sentence is corrected accordingly.**
* * *
technical corrections:

P22 L434: remove '.' at the end of the subtitle

**Re: Removed.**

P29 L589 and L592: '( significant': remove blank

**Re: Removed.**

P31 L637: 'middle- and lower-troposphere': remove '-'

**Re: Removed.**

**References:**

**Park, M., Randel, W. J., Emmons, L. K. and Livesey, N. J.: Transport pathways of carbon monoxide in the Asian summer monsoon diagnosed from Model of Ozon and Related Tracers (MOZART), J. Geophys. Res., 144, D08303,**

doi:10.1029/2008JD010621, 2009.

Xiao, Y., Jacob, D. J., and Turquety, S.: Atmospheric acetylene and its relationship with CO as an indicator of air mass age, J. Geophys. Res., 112,

D12305, doi:10.1029/2006JD008268, 2007.